# TokenSeek: Memory Efficient Fine Tuning via Instance-Aware Token Ditching

**Runjia Zeng[1], Qifan Wang[2], Qiang Guan[3], Ruixiang Tang[4],**
**Lifu Huang[5], Zhenting Wang[6], Xueling Zhang[1], Cheng Han[7], Dongfang Liu[1†]**

[1]Rochester Institute of Technology    [2]Meta AI    [3]Kent State University    [4]Rutgers University
[5]UC Davis    [6]Accenture    [7]University of Missouri-Kansas City    [†]Corresponding author

## Abstract

Fine-tuning has been regarded as a *de facto* approach for adapting large language models (LLMs) to downstream tasks. However, the high training memory consumption inherited from LLMs makes this process generally inefficient. Among existing memory efficient approaches, activation-related optimization has proven particularly effective, as activations consistently dominate overall memory consumption. Although prior arts offer various activation optimization strategies, they typically adopt a uniform yet inflexible strategy across all instance. This data-agnostic nature ultimately results in ineffective and unstable fine tuning. To solve this problem, we propose TokenSeek, a universal plugin solution that is suitable for various Transformer-based models through instance-aware token seeking and ditching. TokenSeek achieves significant fine-tuning memory savings (*e.g.*, requiring only 2.8 GB, 14.8% of the original memory on Llama3.2 1B) with on-par or even superior performance. Furthermore, our interpretable token seeking process reveals the underlying factors behind its effectiveness, offering valuable insights for future research on token efficiency fine-tuning. Homepage: runjia.tech/iclr_tokenseek.

## 1 Introduction

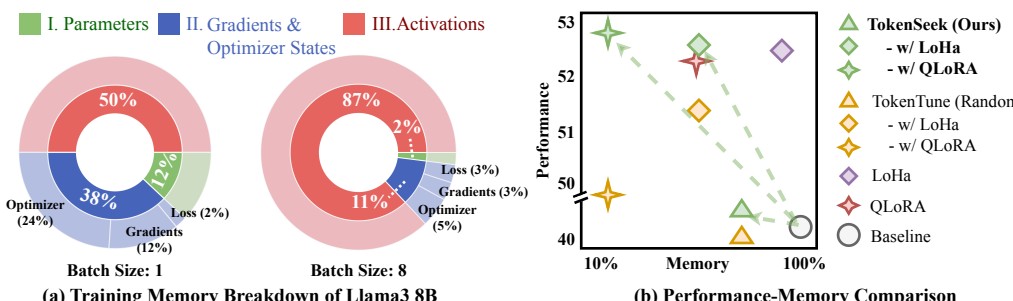

Figure 1: **Motivation behind TokenSeek and its preliminary comparison**. (a) Breakdown of training memory under different batch side settings, revealing that activations are the primary bottleneck in training memory consumption. (b) Effective and efficient TokenSeek (ours) *vs.* concurrent arts in performance and memory consumption on Llama3.2 1B (detailed results in Tab. 1)

"Pretrain-then-Finetune" paradigm (Liu et al., 2024a; Yang et al., 2024; Grattafiori et al., 2024) has been regarded as a *de facto* approach for downstream task adaptation, leveraging the knowledge acquired during pre-training. However, fine tuning large language models (LLMs) still imposes significant memory demands arising from multiple components (Zhang & Su, 2025; Rajbhandari et al., 2020) as showin in Fig. 1 (a), including the model **I.** parameters, **II.** gradients and optimizer states, and intermediate **III.** activations. Current works optimize training memory usage by targeting different components. Parameter-Efficient Fine-Tuning (PEFT) (Zeng et al., 2024; Han et al., 2023; 2024) reduces the number of tunable parameters required for adapting large models (component I). Optimizer-Efficient Fine-Tuning (Rajbhandari et al., 2020; Anil et al., 2019) focuses on partitioning or improving the efficiency of gradient updates and optimizer states to alleviate the training memory burden (component II). Memory-Efficient Fine-Tuning (MEFT) (Simoulin et al., 2024; Dettmers et al., 2023) , on the other hand, improves memory efficiency by recomputing, compressing, or

eliminating activation-related memory costs (component III). Among the three paradigms that address the memory challenge from different perspectives, MEFT stands out as a more effective one. The reason is that activations consistently dominate memory consumption (*e.g.*, 87% in Llama3 8B as shown in Tab. 1 (a), and 60GB of activations for GPT-2 1.5B (Rajbhandari et al., 2020)), making them a critical bottleneck in the memory efficiency of training deep models (Zhang & Su, 2025).

However, existing MEFT methods generally unaware of or ignore the abundant information contained in the fine tuning training instances, *i.e.*, they operate as **data-agnostic** optimizations. Previous works are predominantly model-oriented optimizations (see §2) — they adopt a uniform efficiency strategy across all instances, without accounting for the rich variability inherent within each individual instance. This results in a lack of fine-grained control over memory reduction at the instance level, leading to ineffective (see Fig. 1 (b) and §4.2) and unstable fine-tuning (see §4.3). Naturally, two key challenges arise on the path toward instance-aware activation efficient optimization: I. *how to identify* the salient tokens that represent the key information of each instance (solved through §3.2.1); and II. *how to leverage* them to achieve effective and stable memory optimization (solved through §3.2.2).

In light of this view, we introduce TOKENSEEK, a universally applicable plugin designed to achieve a win-win of performance and memory efficiency without altering their inherent architecture under the "Pretrain-then-Finetune" pradigm. In order to kill two birds with one stone, our approach can be unfolded into two aspects to respectively tackle the challenges above: ❶ **Instance-Aware Token Seeking**. TOKENSEEK first leverages context and gradient information at the token level to evaluate and score individual tokens, selectively retaining more informative ones to mitigate performance degradation and fluctuation. ❷ **Efficient Token Ditching**. TOKENSEEK then significantly decreases the memory footprint for activations by updating model parameters exclusively on selected tokens, thereby ditching the gradients of the others and thus eliminating these activations. Our method facilitates an adaptive, instance-aware activation optimization without compromising performance and stability (see more discussions in §S7). Our key contributions include:

- **Significant Memory Reduction**: Benefit from the potent instance awareness, TOKENSEEK can achieve substantial memory savings with only 10% tokens (*i.e.*, 65.7% maximum memory reduction on Llama3.2 1B, see §4.2) while maintaining competitive performance (*i.e.*, 41.13 *vs.* 40.82). Our approach can further significantly surpass full token fine-tuning with only 14.8% memory consumption under the QLoRA settings (*i.e.*, 52.61 *vs.* 40.82 shown in Fig. 1 (b)).

- **Generalizable Solution**: Attributed to its architecture-agnostic design, our method generalizes well across various Transformer-based models (*i.e.*, Qwen-0.5B, Llama-1B and Llama-3B) and can be seamlessly integrated with other PEFT techniques (*i.e.*, LoHa and QLoRA) to embrace both performance effectiveness and memory efficiency (see Tab. 1).

- **Interpretable Token Seeking**: We provide a comprehensive analysis (see §4.3) of how token-level ditching influences the fine-tuning process, achieving significant memory reductions through our proposed transparent and explainable token selection strategy (see §3.2 and §4.3).

## 2 RELATED WORK

### 2.1 MEMORY-EFFICIENT FINE-TUNING

MEPT (Simoulin et al., 2023; Vucetic et al., 2022; Ryu et al., 2024; Zhang et al., 2023; Zhao et al., 2024; Ardakani et al., 2023) directly tragets on reducing memory footprints during fine tuning. It can be broadly categorized into the *recomputation*, *compression*, and *reversible network* paradigms.

*Recomputation.* The core idea of recomputation methods (Korthikanti et al., 2023; Chen et al., 2024b; Tang et al., 2024) is to recompute certain operations instead of storing all intermediate activations — a technique also known as gradient checkpointing. (Chen et al., 2016) first applied this idea to deep neural networks, proposing a method that stores only a subset of activations and recomputes others during the backward pass, achieving sublinear memory cost. Subsequent improvements optimized the checkpointing schedule (Jain et al., 2020), introduced dynamic runtime strategies (Kirisame et al., 2020), and combined offloading with recomputation (Rajbhandari et al., 2020). *Compression.* Compression methods (Yi et al., 2024; Yang et al., 2025; Leconte et al., 2024) focus on reducing the size of the model states, optimizer states, gradients, and activations, which can further divided into methods using sparsified and quantized representations. Specifically, sparsity methods include LoRA (Hu et al., 2022a), which freezes pre-trained weights and trains low-rank adapters, and diff pruning

(Guo et al., 2020), which learns sparse task-specific updates. Recently, TokenTune (Simoulin et al., 2024) reveals the feasibility of token pruning during backpropagation to further reduces memory by selectively dropping token activations. Quantization-based methods, on the other hand, use lower numerical precision to minimize memory usage. Mixed-precision training (Micikevicius et al., 2017) with FP16 or BF16 became standard, and further advancements introduced 8-bit optimizer quantization. QLoRA (Dettmers et al., 2023) extends this by applying 4-bit quantization to model weights during fine-tuning. *Reversible Networks.* Reversible network designs eliminate the need to cache activations during training by reconstructing them from outputs. RevNets (Gomez et al., 2017) demonstrated reversible residual blocks for image models, while Reformer (Kitaev et al., 2020) extended this idea to Transformers with reversible layers. Recent methods adapt reversible computation to fine-tuning pre-trained models by inserting reversible adapters (Liao et al., 2023b), significantly reducing activation memory without modifying the pre-trained weights.

TOKENSEEK, a sparsified gradient updating method under *compression* paradigm , leverages both context and gradient information in each sample to enable instance-aware activation sparsification with performance on par with dense models, bridging the performance gap.

## 2.2 PARAMETER-EFFICIENT FINE-TUNING

PEFT (Hu et al., 2022b; Aghajanyan et al., 2020; Yang et al., 2023; Huang et al., 2023; Zadouri et al., 2023) aims to optimize model parameter usage and thus can reduce memory consumption to varying degrees. It can be generally categorized into four paradigms (see more in §S7).

*Partial Tuning* methods (Lawton et al., 2023; Xu et al., 2021) update only a subset of the backbone model parameters using weight masking or partial tuning strategies. A common strategy is to fine-tune only the final few layers or sorely the output head. However, its simplistic strategy may directly result in performance degradation, motivating further research into targeted masked tuning approaches (Sung et al., 2021; Chen et al., 2024a; Liao et al., 2023a). *Additional Tuning* methods introduce a small number of new parameters to a frozen pre-trained model, fine-tuning only the added modules. These methods can be further categorized into adapter-based (Houlsby et al., 2019; Pfeiffer et al., 2020; Wang et al., 2022a) and prompt-based approaches (Jia et al., 2022; Wang et al., 2024; 2023), which inject lightweight learnable modules into the model architecture or the model input, respectively. *Reparameterized Tuning* methods reparameterize the model updates in a low-dimensional subspace (Liu et al., 2025b), leveraging the low intrinsic dimensionality of LLMs. LoRA (Hu et al., 2022a) learns low-rank matrices to model weight updates without modifying the original weights. Subsequent works have extended LoRA to alternative reparameterization variants (Hyeon-Woo et al., 2021) and incorporated quantization techniques for additional memory savings (Dettmers et al., 2023). *Hybrid Tuning* methods (He et al., 2021; Zhang et al., 2024) combine multiple PEFT strategies, aiming to unify their advantages. UniPELT (Mao et al., 2021) stands out as a representative method that jointly incorporates adapters, prefix tuning, and LoRA-style low-rank updates as submodules, and learns to activate those best suited to the current task via gating.

While PEFT methods primarily focus on parameter efficiency (*i.e.*, reducing component I storage), their impact on overall memory efficiency is limited (*e.g.*, the activation memory of most PEFT methods remains over 75% of that in full fine-tuning, even with less than 1% trainable parameters (Liao et al., 2023b)). Leveraging our architecture-agnostic design, TOKENSEEK can be seamlessly integrated with PEFT methods, further embracing both parameter and memory efficiency (see Tab. 1).

## 3 METHODOLOGY

In §3.1, we first analyze activations, the primary bottleneck in training memory, from two perspectives: (i) why storing activations is necessary, and (ii) why they incur large memory consumption in LLMs. Our method, TOKENSEEK, is presented in §3.2, which is decomposed into two key components: instance-aware token seeking and efficient token ditching. The overall framework is shown in Fig. 2.

### 3.1 PRELIMINARY

**The Necessity of Storing Activations**. Given a multilayer deep neural network, we first analyse the memory consumption of activations, in which the transformation and nonlinear activation at layer $l$ are defined by $a^{(l)} = z^{(l-1)}W^{(l)} + b^{(l)}$ and $z^{(l)} = \sigma\big(a^{(l)}\big)$, respectively. Here, the weight matrix $W^{(l)}$ projects the output $z^{(l-1)}$ of the previous layer into the current layer's pre-activation $a^{(l)}$, to which we add the bias $b$ before applying the activation function $\sigma$. By extending to deeper layers

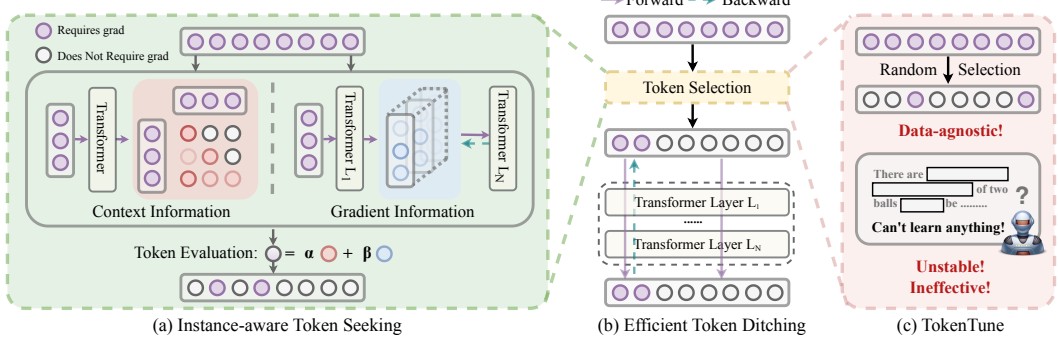

Figure 2: **Overview of TOKENSEEK (ours)** *vs.* **TOKENTUNE frameworks.** (a) Instance-aware token seeking using context and gradient information (see §3.2.1 and Eq. 5). (b) Efficient token ditching (see §3.2.2). (c) TOKENTUNE for random token selection (see analysis in Tab. 1 and §4.3).

and applying the differentiation rules along with the chain rule, we can decompose the gradient with respect to the weight in the first layer in simplicity as:

$$\frac{\partial \mathcal{L}}{\partial W^{(1)}} = \frac{\partial \mathcal{L}}{\partial z^{(L)}} \Big( \prod_{\ell=2}^{L} \boxed{\frac{\partial z^{(\ell)}}{\partial z^{(\ell-1)}}} \Big) \frac{\partial z^{(1)}}{\partial W^{(1)}}. \tag{1}$$

$$\boxed{\frac{\partial z^{(l)}}{\partial z^{(l-1)}}} = \frac{\partial z^{(l)}}{\partial a^{(l)}} \frac{\partial a^{(l)}}{\partial z^{(l-1)}} = \sigma'\big(a^{(l)}\big) W^{(l)}. \tag{2}$$

The computation of the back-prop term $\sigma'\big(a^{(l)}\big) W^{(l)}$ requires the intermediate value $a^{(l)}$ to further evaluate $\sigma'\big(a^{(l)}\big)$. By caching each pre-activation $a^{(l)}$ during the forward pass, the model can avoid recomputing to obtain these intermediates, thereby efficiently forming the full chain of derivatives.

**The Reason of Large Activations**. Current Transformer-based LLMs follow this rule to store activations during backpropagation. Taking DeepSeek-v3 (Liu et al., 2024a; Zhang & Su, 2025) as an example, the activations in each layer have a space complexity of $\mathcal{O}(Bn_h s^2 + BsH)$, where $B$ is the batch size, $n_h$ is the number of attention heads, $s$ is the sequence length, and $H$ is the hidden dimension. The space complexity required for activations significantly outweighs that of the weights ($Bn_h s^2 + BsH \gg H^2$ given $B = 1$, $n_h = 128$, $s = 4096$ and $H = 7168$ (Zhang & Su, 2025)).

## 3.2 TOKENSEEK

### 3.2.1 INSTANCE-AWARE TOKEN SEEKING

The key insight behind TOKENSEEK is that not all training tokens within LLMs contribute equally to model fine-tuning, known as token redundancy. Token redundancy has long been recognized as a fundamental challenge to LLM efficiency (Hou et al., 2022), drawing increasing research attention across various domains, including efficient chain-of-thought reasoning (Xia et al., 2025) and prompt optimization (Li et al., 2023). This observation greatly inspires us to explore the potential of memory-efficient fine-tuning by reducing token redundancy. Then the critical problem turns to determine the importances of each token. Here, we propose a comprehensive evaluation of tokens by leveraging both **context** and **gradient information** captured within Transformer blocks (see Fig. 2 (a)).

➤ **Context Information**. Most LLMs are built upon the Transformer architecture, which fundamentally relies on the attention mechanism (Vaswani et al., 2017). The attention maps in decoder layers directly reflect the importance of each token in context, thereby guiding and shaping the transformation process. More specifically, given input embeddings $t \in \mathbb{R}^{n \times d}$, we first project them into queries and keys with learnable matrices $W^Q, W^K \in \mathbb{R}^{d \times d}$ to obtain $\mathbf{Q} = tW^Q$, $\mathbf{K} = tW^K$. We then form the attention scores and apply a causal mask for language modeling, computing $\mathbf{A} = \mathrm{softmax}\Big(\mathrm{mask}\big(\mathbf{Q}\mathbf{K}^\top/\sqrt{d_k}\big)\Big)$. Each entry $\mathbf{A}_{ij}$ denotes the attention weight from token $i$ to token $j$. Owing to the row-wise softmax normalization, each row $\mathbf{A}_{i:}$ forms a probability distribution over all tokens, capturing how much attention token $i$ allocates to others. Conversely, each column

$\mathbf{A}_{:j}$ reflects the cumulative attention received by a token $j$ from all other tokens in the sequence. In this way, attention mechanism provides an intuitive and direct measure of a token's importance within a given instance. The context importance of each token (Singh et al., 2024; Liao et al., 2025; Kong et al., 2023) is computed as:

$$I_1(t_j) = \sum_{i=1}^{n} \mathbf{A}_{ij}. \tag{3}$$

➤ **Gradient Information**. However, context-based evaluation above only reflects the importance of a token within a given instance and does not necessarily indicate its contribution to model fine-tuning. Therefore, in order to better quantify the contribution of each token during the fine tuning, we further assess token importance by examining the gradient magnitude of the loss $w.r.t.$ the activations. This idea is inspired by (Jain & Wallace, 2019), which shows that attention weights are often uncorrelated with gradient-based measures of feature importance. The study positions gradient-based attribution as a more reliable yardstick of "true" token importance, highlighting that gradient-based saliency can substantially differ from attention-based explanations (see more discussion in §4.3). Given the gradient matrix $\mathbf{G} = \left[\partial\mathcal{L}/\partial z^{(L-1)}\right] \in \mathbb{R}^{n \times d}$ for the activations in the penultimate layer (*i.e.*, the input to the final decoder layer) computed during backpropagation, the gradient-based importance of each token is computed by summing the gradient magnitudes across the hidden dimension as:

$$I_2(t_j) = \text{Accumulate}\left[\frac{\partial\mathcal{L}}{\partial z^{(L-1)}}\right], \quad \text{Accumulate}[\cdot] = \sum_{k=1}^{d} \mathbf{G}_{jk}. \tag{4}$$

➤ **Token Evaluation**. To obtain a comprehensive evaluation of token importance, we integrate both context and gradient information, weighted by scalars $\alpha$ and $\beta$, respectively as:

$$I(t_j) = \alpha \log\left[I_1(t_j)\right] + \beta \, \text{Norm}\left[I_2(t_j)\right], \tag{5}$$

where we apply a log-like transformation to address the long-tail distribution of contextual importance scores, and use min-max normalization to scale the gradient-based importance scores to a comparable range (see more discussions in §4.3). By incorporating both information, our method is able to robustly evaluate tokens within each instance (distinct from random selection in Fig. 2 (c)), leading to more effective and stable fine-tuning (see §4.2 and §4.3). This enables us to select the tokens that contribute most to model fine tuning in the subsequent memory-efficient token ditching (see §3.2.2).

### 3.2.2 EFFICIENT TOKEN DITCHING

To improve training memory efficiency, we propose ditching the gradients of less informative tokens from the dataset, and fine-tuning LLMs using only the selected tokens. Following (Simoulin et al., 2024), by backpropagating the loss through the selected tokens in $t$ only, and ditching the gradient computation of unselected tokens $\bar{t}$ in Eq. 2 (see Fig. 2 (b)) after regrouping them, we have:

$$\boxed{\frac{\partial z^{(l)}}{\partial z^{(l-1)}}} = \left[\,\sigma'\!\left(a_t^{(l)}\right), \sigma'\!\left(a_{\bar{t}}^{(l)}\right)\,\right] W^{(l)} = \left[\,\sigma'\!\left(a_t^{(l)}\right), \; 0 \,\right] W^{(l)}. \tag{6}$$

Based on Eq. 6, we only need to cache $a_t^{(l)}$ to apply the chain rule, rather than storing the full activation $a^{(l)}$. We provide a detailed discussion in §S2 and §S7.1.

### 3.2.3 ANALYSIS AND DISCUSSION

As shown in §3.2.1, in contrast to other token importance evaluation strategies that require additional annotations or auxiliary networks (Xia et al., 2025), TOKENSEEK only requires a forward pass (*e.g.*, Llama3 8B requires only 13.3% of the training memory during inference under the FP8 setting) and a partial backward pass (*i.e.*, freeze all layers except the output head and the final decoder block) to assess token importance, resulting in *simplicity*. Presented in §3.2.2, tuning only 10% of tokens theoretically requires just ~1% of the activation memory, based on the space complexity analysis in §3.1, resulting in a highly *efficient* fine-tuning process. Beside enjoying the appealing characteristics of *simplicity* and *efficiency*, TOKENSEEK also has merits in *generality* and *interpretability*. For *generality*, our method relies solely on context and gradient information, making it architecture-agnostic and broadly applicable to a more wide range of Transformer-based models (see Tab.1). For *interpretability*, both context and gradient information provide intuitive and direct evaluations of token importance, thereby further enhancing the interpretability of our evaluation process (see §4.3).

Table 1: **Few-shot evaluation on question-answering benchmarks**. This includes ARC (25-shot) , MMLU (5-shot) , HellaSwag (10-shot) , TruthfulQA (0-shot) , and WinoGrande (0-shot). We report the average accuracy on five MMLU ethics tasks and WinoGrande, the normed accuracy on ARC and HellaSwag, and the MC2 score on TruthfulQA. The number reported in [·] is the "Tuned/Total" parameters in each setting. The same training settings are highlighted in blue, red, and orange for full-parameter tuning, LoHa, and QLoRA, respectively. Relative memory consumption percentages compared with the settting of full token tuning are transformed and reported in each scale. Same for Tab.2. We highlight the best average performance and memory savings in **bold**. For TOKENTUNE and TOKENSEEK, only 10% of the input tokens are selected for gradient computation.

| Method | Ave. Mem. | Max. Mem. | ARC | Hella Swag | MMLU | Truthful QA | Wino Grande | Average Score |
|---|---|---|---|---|---|---|---|---|
| **Qwen2.5** (0.5B) | | | | | | | | |
| Full Parameter/Token Tuning | 100% | 100% | 34.89 | 51.70 | 59.20 | 39.86 | 56.51 | 48.43 |
| - w/ TOKENTUNE (Random) | 48.3% | 25.6% | 25.26 | 25.78 | 51.07 | 49.93 | 47.36 | 39.88 |
| - w/ **TOKENSEEK (Ours)** | 48.3% | 25.6% | 25.17 | 25.52 | 58.14 | 50.13 | 50.75 | 41.94 |
| IA3 (Liu et al., 2022) | 84.3% | 72.8% | 34.98 | 51.66 | 56.81 | 40.08 | 56.51 | 48.01 |
| LoRA (Hu et al., 2022a) | 81.2% | 71.8% | 34.73 | 51.67 | 56.30 | 41.08 | 56.51 | 48.06 |
| LoKr (Hyeon-Woo et al., 2021) | 91.6% | 79.3% | 35.49 | 51.54 | 58.64 | 39.83 | 55.88 | 48.28 |
| BOFT (Liu et al., 2024b) | 145.1% | 100.6% | 34.64 | 51.70 | 58.18 | 39.57 | 56.43 | 48.10 |
| Bone (Kang, 2024) | 85.8% | 76.2% | 28.50 | 43.54 | 42.39 | 43.35 | 54.62 | 42.48 |
| LoHa [1.33%] (Hyeon-Woo et al., 2021) | 86.6% | 76.9% | 34.73 | 51.90 | 57.53 | 40.75 | 55.96 | 48.17 |
| - w/ TOKENTUNE (Random) | 39.5% | 22.5% | 23.81 | 26.34 | 57.53 | 50.26 | 47.36 | 41.06 |
| - w/ **TOKENSEEK (Ours)** | 39.5% | 22.5% | 26.54 | 25.96 | 58.14 | 50.26 | 50.51 | 42.28 |
| QLoRA [1.04%] (Dettmers et al., 2023) | 51.7% | 45.6% | 34.64 | 50.10 | 58.05 | 40.41 | 55.09 | 47.66 |
| - w/ TOKENTUNE (Random) | 19.2% | 13.4% | 31.06 | 45.92 | 57.60 | 41.56 | 55.56 | 46.34 |
| - w/ **TOKENSEEK (Ours)** | **19.2%** | **13.4%** | 34.56 | 50.09 | 57.52 | 41.51 | 58.56 | **48.45** |
| **Llama3.2** (1B) | | | | | | | | |
| Full Parameter/Token Tuning | 100% | 100% | 23.72 | 26.11 | 57.53 | 48.68 | 48.07 | 40.82 |
| - w/ TOKENTUNE (Random) | 64.6% | 34.3% | 24.32 | 25.80 | 58.14 | 47.90 | 47.59 | 40.75 |
| - w/ **TOKENSEEK (Ours)** | 64.6% | 34.3% | 23.98 | 25.73 | 58.14 | 48.09 | 49.72 | 41.13 |
| LoHa [0.63%] | 92.3% | 99.4% | 39.25 | 65.93 | 57.60 | 37.87 | 60.77 | 52.28 |
| - w/ TOKENTUNE (Random) | 45.9% | 28.4% | 38.48 | 64.21 | 50.34 | 43.89 | 59.91 | 51.37 |
| - w/ **TOKENSEEK (Ours)** | 45.9% | 28.4% | 38.57 | 65.89 | 58.18 | 39.34 | 60.93 | 52.58 |
| QLoRA [0.52%] | 45.6% | 34.8% | 38.82 | 65.26 | 56.39 | 38.85 | 61.33 | 52.13 |
| - w/ TOKENTUNE (Random) | 14.8% | 14.3% | 39.33 | 62.97 | 41.76 | 41.36 | 60.69 | 49.22 |
| - w/ **TOKENSEEK (Ours)** | **14.8%** | **14.3%** | 39.08 | 65.98 | 58.03 | 38.65 | 61.33 | **52.61** |
| **Llama3.2** (3B) | | | | | | | | |
| Full Parameter/Token Tuning | 100% | 100% | 23.98 | 25.72 | 58.62 | 49.53 | 49.80 | 41.53 |
| - w/ TOKENTUNE (Random) | 73.1% | 39.3% | 24.15 | 25.43 | 57.64 | 50.86 | 47.91 | 41.20 |
| - w/ **TOKENSEEK (Ours)** | 73.1% | 39.3% | 27.30 | 25.96 | 58.14 | 48.65 | 49.72 | 41.95 |
| LoHa [0.47%] | 90.7% | 96.5% | 49.06 | 75.96 | 63.50 | 42.08 | 69.46 | 60.01 |
| - w/ TOKENTUNE (Random) | 49.6% | 30.0% | 50.34 | 75.70 | 56.65 | 43.37 | 69.61 | 59.13 |
| - w/ **TOKENSEEK (Ours)** | 49.6% | 30.0% | 53.24 | 76.81 | 64.31 | 41.75 | 68.98 | **61.02** |
| QLoRA [0.42%] | 33.6% | 26.5% | 51.37 | 75.88 | 63.95 | 42.31 | 68.43 | 60.39 |
| - w/ TOKENTUNE (Random) | 13.3% | 11.1% | 49.91 | 73.04 | 59.91 | 45.25 | 68.43 | 59.31 |
| - w/ **TOKENSEEK (Ours)** | **13.3%** | **11.1%** | 50.00 | 76.30 | 63.43 | 43.37 | 68.98 | 60.42 |

# 4 EXPERIMENT

## 4.1 EXPERIMENTAL SETUP

Under the "Pretrain-then-Finetune" paradigm, pre-trained LLMs are further fine-tuned on specialized datasets to adapt them to domain-specific tasks and improve instruction-following capabilities (Wang et al., 2022b; Taori et al., 2023). In this section, we apply instruction tuning and benchmarking under the few-shot setting. We provide additional experimental details in §S1 and §S7.

**Instruction Tuning.** Following (Simoulin et al., 2024), we fine-tune the Qwen2.5 0.5B (Yang et al., 2024), Llama3.2 1B and 3B (Grattafiori et al., 2024) models using the Open-Platypus dataset (Lee et al., 2023). It comprises 11 open-source instruction datasets. See more details in §S1.7.

**Few-Shot Evaluation.** We assess performance across few-shot benchmarks including MMLU (Hendrycks et al., 2020), ARC (easy and challenge) (Clark et al., 2018), HellaSwag (Zellers et al., 2019), TruthfulQA (Lin et al., 2021), and WinoGrande (Sakaguchi et al., 2021), using the "lm evaluation harness" (Gao et al., 2024). For each task, the model ranks answer options by probability, and the highest one is selected. See more discussions and experiments in §S1, §S3 and §S7.1.

## 4.2 MAIN RESULTS

The main performance and memory comparison results across various models, scales and PEFT settings are shown in Tab. 1, leading to three key observations. *First, Significant Memory Reduction.* TOKENSEEK demonstrates exceptional efficiency in reducing both average and peak memory usage during fine-tuning. Specifically, for the Llama3.2 3B model, peak memory usage is reduced by 60.7% with TOKENSEEK alone (*i.e.*, Llama3.2 3B + TOKENSEEK), and further down to just 11.1% when combined with QLoRA (*i.e.*, Llama3.2 3B + QLoRA + TOKENSEEK), enabling training on a single A100 GPU without triggering OOM issues. Similarly, average memory usage is reduced by 26.9% and 86.7%, respectively. These results highlight TOKENSEEK 's strong capability in achieving extreme memory compression during fine-tuning (more experiments in §4.4). *Second, Competitive or Superior Performance.* Despite memory efficiency, TOKENSEEK maintains competitive or even superior performance compared to full-token tuning: The average score of TOKENSEEK with QLoRA in Qwen 0.5B marginally surpasses the full token tuning baseline (48.45 *vs.* 48.43). In Llama3.2 1B, TOKENSEEK consistently outperforms baseline across all settings (*i.e.*, TOKENSEEK, w/ LoHa, and w/ QLoRA achieve scores of 41.13, 52.58, and 52.61, respectively, compared to 40.82 from full token tuning.). This indicates that the memory compression achieved by TOKENSEEK does not compromise, and may even slightly enhance, model performance (detailed discussions in §4.3). *Third, Generalizable Across Different Scales.* Generally, TOKENSEEK's effectiveness in both memory reduction and performance stability is generalizable across various model scales ranging from 0.5B to 3B. However, we observe a distinct pattern across model scales: TOKENSEEK exhibits performance degradation on Qwen under plain settings, whereas Llama does not. This suggests that TOKENSEEK may be more sensitive to smaller-scale models, likely due to their limited representational capacity. In conclusion, TOKENSEEK demonstrates itself as an universal memory efficient solution, effectively achieving memory efficiency with competitive model accuracy (see more comparison in §S7).

## 4.3 ANALYSIS OF TOKEN SEEKING

**Contributions of Instance-aware Token Seeking.** We further conduct experiments to quantitatively evaluate the impact of instance-aware token seeking (*i.e.*, addressing the drawbacks of data-agnostic optimization: inefficiency and instability) by comparing performance across multiple runs under varying memory settings (*i.e.*, with the ratio of tunable tokens ranging from 10% to 50%) on Llama3.2 1B with QLoRA. As shown in Fig. 3, we observed that TOKENSEEK consistently enhances both *effectiveness* and *stability*. For *effectiveness*, our method (green curve), consistently outperforms the random baseline (purple curve) (Xia et al., 2025) across various memory settings. The accuracy achieved by our approach remains higher at all memory saving levels, clearly demonstrating its effectiveness in optimizing performance under memory constraints. For *stability*, our method showcases superior stability. This is evident from the notably narrower shaded regions (*i.e.*, variance) in Fig. 3, indicating a lower standard deviation compared to the random baseline. The results clearly show TOKENSEEK's capability to maintain higher accuracy with narrower fluctuations, emphasizing its robustness under varying memory constraints.

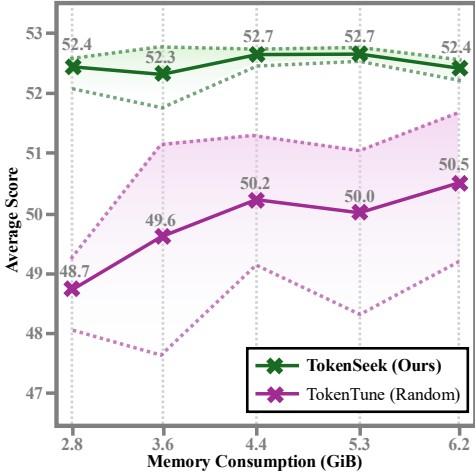

Figure 3: The performance for the Llama3.2 1B with QLoRA setting, where the upper, lower and middle line indicate the maximum, minimum and the average results.

**Study of Interpretability.** One key advantage of TOKENSEEK is the transparency and interpretability of its token seeking and ditching process. To illustrate this, we conduct a case study on a training instance to visualize token evaluation and highlight common patterns (see more visualizations in §S8). TOKENSEEK leverages both contextual and gradient information for token selection. **Context Information**: while token selection varies across instances, our analysis reveals several consistent patterns illustrated in Fig.4 (b). Specifically, *unidirectional* attention enforces causality, resulting in an upper-triangular attention mask. *Diagonal* patterns indicate self-focused attention, emphasizing local context as shown in the blue localized accumulation. *Attention Sink* (Xiao et al., 2024) refers to the tendency of attention to disproportionately concentrate on a single position, effectively acting as a global anchor (evident as the brown line in first column). The context scores

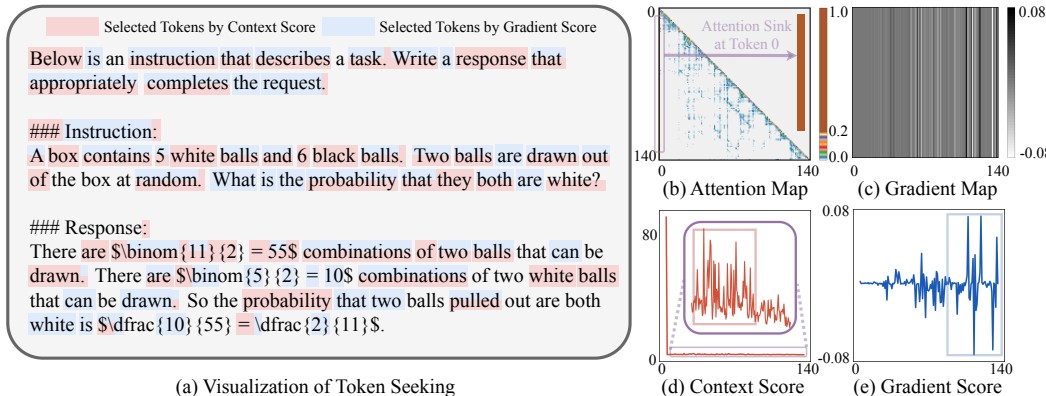

(a) Visualization of Token Seeking

(b) Attention Map  (c) Gradient Map

(d) Context Score  (e) Gradient Score

Figure 4: **Case study of a training instance.** (a) Visualization of the top 50% selected tokens using context and gradient information, highlighted in red and blue, respectively. (b) Average attention map from the final layer. (c) Accumulated gradient map of activations in the penultimate layer. (d) Context importance scores obtained by column-wise accumulation. (e) Gradient importance scores obtained by summing across the hidden dimension. Additional visualizations are provided in §S8.

exhibit a long-tail distribution caused by the attention sink effect, with higher scores concentrated in earlier positions—amplified by the causal mask (see zoom-in area of Fig.4 (d)). This results in a preference for earlier tokens, as reflected in the red-highlighted regions of Fig. 4 (a), which retain semantically meaningful tokens (e.g., those related to mathematical learning) while filtering out less informative ones, such as definite articles and prepositions. **Gradient Information**: to enhance interpretability, we aggregate the gradient map along the hidden dimension, producing $\mathbf{G}' \in \mathbb{R}^{n \times 1}$, which is visualized as a color map in Fig.4 (c). As illustrated in Fig.4 (e), the gradient information predominantly focuses on later positions—typically corresponding to the "Response" portion of a training instance (highlighted in blue in Fig. 4 (a)), underscoring the importance of learning the answer generation process. In summary, our findings highlight the interpretability of TOKENSEEK, showing that context and gradient information exhibit distinct but complementary patterns (see more in §S7). Their integration enables a more comprehensive and robust approach to token evaluation.

**Study of Optimization**. We further investigate the reason of superior performance under the PEFT settings and discuss the potential impact of the percentage of tunable tokens during fine tuning. By comparing different tunable token ratios within each group (*i.e.*, lighter *vs*. darker lines) and across different tuning strategies (*i.e.*, different colored lines), we have two key observations. **i) TOKENSEEK favors PEFT**: Considering PEFT methods (*i.e.*, LoHa and QLoRA), we observe that full parameter tuning yields lower training loss, which may indicate potential overfitting (*i.e.*, lower training loss but poorer downstream performance). In contrast, PEFT methods, which update only a subset of parameters, are less prone to overfitting and therefore remain more robust and better suited for token ditching (*i.e.*, 47.66 on QLoRA *vs*. 48.85 on QLoRA + TOKENSEEK). **ii) Tokens Contribute Fine Tuning**: As more tunable tokens are incorporated into fine-tuning (*i.e.*, from 10% to 50%), we consistently observe lower optimization loss, which indicates that fine tuning is sensitive to training data volume. Low-quality

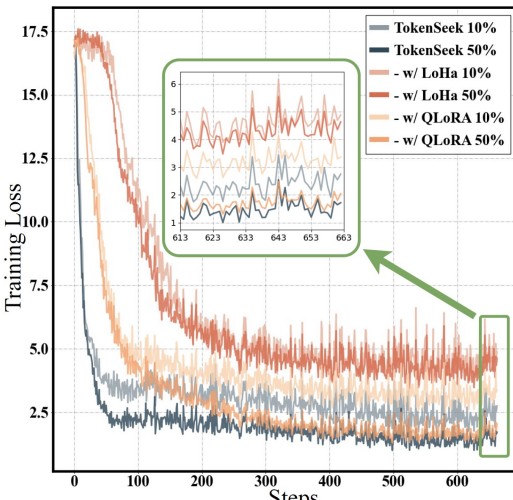

Figure 5: **Training loss curves of six settings on Qwen2.5 0.5B.** Blue, red, and orange lines represent full parameter, LoHa, and QLoRA tuning, respectively. Lighter lines in each group indicate 10% token tuning, while darker lines indicate 50%. Detailed performance and memory usage results are provided in Tab. 1 and 2.

token selection under extremely limited token training may thus lead to optimization collapse and performance degradation, emphasizing the need for our instance-aware token seeking approach.

Table 2: **Ablation study on token evaluation in Eq. 5**, analyzing the impact of scalar weighting and threshold selection on both memory consumption and model performance (See more in §S7)

| Settings | Ave. Mem. | Max. Mem. | MMLU | ARC | Hella Swag | Truthful QA | Wino Grande | Average Score |
|---|---|---|---|---|---|---|---|---|
| \multicolumn{9}{c}{**Sensitivity to Scalar** (Qwen2.5 0.5B with QLoRA)} |
| $\alpha$=1, $\beta$=0 | | | 34.56 | 50.09 | 57.52 | 41.51 | 58.56 | 48.45 |
| $\alpha$=0, $\beta$=1 | | | 30.72 | 44.20 | 57.62 | 43.98 | 55.41 | 46.39 |
| $\alpha$=5, $\beta$=5 | 19.2% | 13.4% | 35.15 | 50.20 | 58.49 | 41.48 | 57.93 | **48.65** |
| $\alpha$=3, $\beta$=7 | | | 34.64 | 48.77 | 58.33 | 42.18 | 57.46 | 48.28 |
| $\alpha$=7, $\beta$=3 | | | 35.58 | 50.10 | 58.59 | 41.13 | 57.22 | 48.53 |
| \multicolumn{9}{c}{**Ratio of Tunable Token** (Llama3.2 1B with QLoRA)} |
| 100% | 100% | 100% | 23.72 | 26.11 | 57.53 | 48.68 | 48.07 | 40.82 |
| 50% | 32.6% | 53.1% | 39.42 | 65.34 | 55.00 | 40.53 | 61.01 | 52.26 |
| 40% | 28.0% | 29.9% | 39.42 | 65.51 | 56.92 | 39.89 | 61.56 | 52.66 |
| 30% | 23.2% | 23.7% | 40.27 | 65.60 | 54.63 | 41.46 | 61.80 | **52.75** |
| 20% | 18.8% | 18.6% | 39.42 | 65.73 | 53.24 | 39.86 | 60.77 | 51.80 |
| 10% | 14.8% | 14.3% | 39.08 | 65.98 | 58.03 | 38.65 | 61.33 | 52.61 |

## 4.4 ABLATION STUDY

**Sensitivity to Scalar**. As stated in Eq. 5, the scalars $\alpha$ and $\beta$ determine the relative emphasis placed on context and gradient information during token evaluation, respectively. As shown in Tab. 2, configurations that balance both information (*e.g.*, [5, 5]) achieve higher performance compared to those relying on a single evaluation (*e.g.*, [0, 1]) . Notably, TOKENSEEK performance remains stable across various settings, indicating low sensitivity within certain range.
**Ratio of Tunable Token**. We then investigate the ratio of tunable tokens *w.r.t.* model performance, particularly under scenarios with extremely limited gradient tokens. Tab. 2 shows that TOKENSEEK maintains stable performance across a range of ratios (*i.e.*, from 51.80 to 52.75), while memory usage decreases substantially as the ratio reduces (*i.e.*, from 32.6% to 14.8%). Given the observed stability, we suggest a default ratio of 10% for overall efficiency.
**GPU Memory Impact**. The motivation behind memory-efficient fine-tuning stems from the mismatch between limited GPU memory (*e.g.*, 40GB

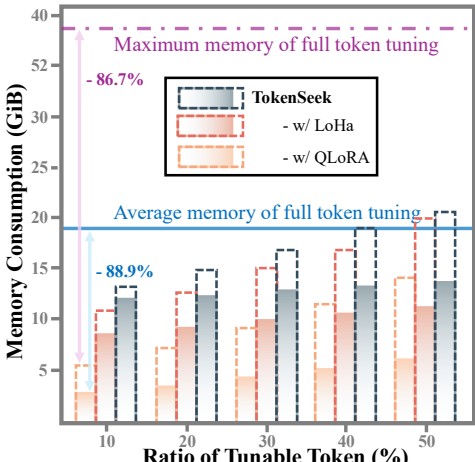

Figure 6: **Training memory under different settings and token ratio selections.** Bars represent average memory usage, while dashed lines indicate peak memory consumption.

on A100, 24GB on RTX 4090) and the growing size of LLMs. In this context, peak memory determines whether a model can be fine-tuned on a single GPU without encountering OOM issues. Benefit from our model-agnostic design, TOKENSEEK provides faithfully cumulative memory savings when combined with PEFT methods. For example, when integrated with QLoRA, the peak memory usage ranges from 14.2 GiB to as low as 5.5 GiB depending on the ratio selected for tunable tokens — substantially lower than the 38.8 GiB required by full token tuning.

## 5 CONCLUSION

We propose TOKENSEEK, a universal plugin solution for effective and stable Transformer-based memory efficient fine tuning. It has merits in: **i)** significant memory reduction via token ditching without sacrificing performance; **ii)** strong generalizability across various LLMs and compatibility with existing PEFT methods; and **iii)** interpretable instance-aware seeking for effective and stable fine tuning. As a whole, we conclude that the outcomes elucidated in this paper impart essential understandings and thus necessitate further exploration within the field of MEFT.

## 6 ACKNOWLEDGEMENTS

This research was supported by the National Science Foundation under Grant No. 2450068. This work used NCSA Delta GPU through allocation CIS250460 from the Advanced Cyberinfrastructure Coordination Ecosystem: Services & Support (ACCESS) program, which is supported by U.S National Science Foundation grants No. 2138259, No. 2138286, No. 2138307, No. 2137603, and No. 2138296. We also gratefully acknowledge the support of RIT Research Computing (of Technology, 2025).

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

## SUMMARY OF THE APPENDIX

This appendix contains additional experimental results and discussions of our ICLR 2026 submission: TOKENSEEK: *Memory Efficient Fine Tuning via Instance-Aware Token Ditching*, organized as follows:

- §S1 provides **additional implementation details of TOKENSEEK**, complementing the overall methodology and results presented in the main paper.
- §S2 offers a **detailed analysis of efficient token ditching**, expanding on the motivations and complexity analysis discussed in the main paper.
- §S3 presents **additional experiments of TOKENSEEK on larger LLMs**, extending the smaller-scale evaluations included in the main paper.
- §S4 shows related **asset license and consent** to our work.
- §S5 claims **reproducibility** of our approach.
- §S6 discusses the **social impact** of our research.
- §S7 adds more **discussions**, and points out potential directions of our **future work**.
- §S8 includes further **visualization results**, covering case studies, attention maps, and gradient score distributions.

## S1 IMPLEMENTATION DETAILS

### S1.1 INSTRUCTION TEMPLATE

For instruction tuning of large LLMs, we apply the Alpaca (Taori et al., 2023) prompt template without incorporating step-by-step reasoning following (Simoulin et al., 2024), as shown below.

```
"Below is an instruction that describes a task, paired with an input that provides
further context. Write a response that appropriately completes the request.
### Instruction: {instruction}
### Input: {input}
### Response:
"
```

### S1.2 TRAINING AND EVALUATION DATA

For instruction tuning, we fine-tuned Qwen and Llama on 21,221 samples from the Open-Platypus dataset (Lee et al., 2023). Although Open-Platypus comprises 11 open-source datasets, we excluded two—*leetcode-solutions-python-testgen-gpt4* and *airoboros-gpt4-1.4.1*—as they contain outputs generated by GPT models (Achiam et al., 2023). We used the remaining 9 datasets for fine-tuning, following (Simoulin et al., 2024).

### S1.3 TRAINING SETTINGS

We do not use any checkpointing, ZeRO or offloading techniques. Below are other training settings details.

```
bf16: true
fp16: false
fp16_opt_level: "O1"
lr_scheduler_type: "cosine"
warmup_steps: 100
weight_decay: 0.01
optim: "adamw_torch"
```

For QLoRA and LoHa, we use the below same configuration in all experiments.

```
alpha:16
dropout:0.05
r:8
```

For other baselines such as IA3 and BOFT, we use the default PEFT configuration across all runs.

Regarding the resulting peak/average memory, we would be happy to provide the original numbers for full parameter/token tuning in each setting for more comprehensive understanding. Other baseline memory usage can be calculated using the original numbers below and their relative memory percentages reported in Tab. 1.

Table S1: Memory usage summary for Qwen2.5 and Llama3.2 models under full parameter/token tuning.

| Model | Average Memory (MB) | Maximum Memory (MB) |
|---|---|---|
| Qwen2.5 (0.5B) | 12,070 | 27,300 |
| Llama3.2 (1B) | 19,462 | 39,688 |
| Llama3.2 (3B) | 40,100 | 83,927 |

## S1.4 HYPERPARAMETER SETTINGS

Given the hyperparameters $\alpha$ and $\beta$ introduced in Eq. 5, we perform a linear search over the set $\{[1,0], [0,1], [5,5]\}$ for each LLM. The optimal settings are: $[1,0]$ for Qwen2.5 0.5B and Llama3.2 1B, and $[5,5]$ for Llama3.2 3B. A linear search over $\{[1,0], [0,1], [5,5]\}$ reveals that smaller models (Qwen2.5 0.5B, Llama3.2 1B) benefit from context-only information, while the larger Llama3.2 3B performs the best when combining both context and gradient information. This suggests that larger models may better utilize multi-information token evaluation.

## S1.5 STANDARD ERROR FOR MAIN RESULTS

We report standard errors in Tab. S2 for the main evaluation results presented in Tab.1. It is important to note that these values are derived from the "lm evaluation harness" (Gao et al., 2024), which yields consistent standard error across repeated evaluations on the same task and dataset. Consequently, all PEFT baselines (*e.g.*, LoHa, QLoRA) and TOKENTUNE variants share similar standard errors for each benchmark under the same task setting. This consistency ensures a fair and controlled comparison across methods. As shown, the average standard errors across all tasks remain low (*e.g.*, $\sim$1.0), indicating stable performance estimates. Therefore, the conclusions drawn from accuracy improvements and memory savings remain robust under the evaluation framework, further supporting the reliability of TOKENSEEK 's performance gains.

Table S2: Standard error for the main results in Tab. 1.

| Method | ARC | Hella Swag | MMLU | Truthful QA | Wino Grande | Average |
|---|---|---|---|---|---|---|
| **Qwen2.5** (0.5B) | | | | | | |
| **TOKENSEEK (Ours)** | 1.216 | 0.438 | 0.339 | 1.640 | 1.405 | 1.008 |
| - w/ LoHa | 1.252 | 0.439 | 0.338 | 1.631 | 1.405 | 1.013 |
| - w/ QLoRA | 1.395 | 0.499 | 0.337 | 1.436 | 1.387 | 1.011 |
| **Llama3.2** (1B) | | | | | | |
| **TOKENSEEK (Ours)** | 1.245 | 0.437 | 0.338 | 1.623 | 1.405 | 1.009 |
| - w/ LoHa | 1.433 | 0.473 | 0.345 | 1.421 | 1.367 | 1.008 |
| - w/ QLoRA | 1.434 | 0.477 | 0.342 | 1.455 | 1.372 | 1.016 |
| **Llama3.2** (3B) | | | | | | |
| **TOKENSEEK (Ours)** | 1.237 | 0.442 | 0.337 | 1.613 | 1.404 | 1.007 |
| - w/ LoHa | 1.461 | 0.427 | 0.330 | 1.474 | 1.287 | 0.996 |
| - w/ QLoRA | 1.450 | 0.432 | 0.331 | 1.497 | 1.290 | 1.000 |

## S1.6 TRAINING TIME FOR MAIN RESULTS

We further analyze the training time (measured in GPU hours on one NVIDIA A100-40GB GPU) to assess the computational efficiency of TOKENSEEK compared to other methods. Notably, TO-

KENSEEK exhibits similar training time overhead to TOKENTUNE across all model scales and PEFT settings. Specifically, both TOKENSEEK and TOKENTUNE incur a modest increase of approximately 11–15% in GPU hours compared to baseline full-token tuning. For example, on Qwen2.5 0.5B, the baseline takes 0.43 GPU hours, while TOKENSEEK requires 0.49; similarly, on Llama3.2 1B, the baseline uses 0.35 hours $vs.$ 0.39 hours for TOKENSEEK. This overhead is likely attributed to the irregularity introduced by selective gradient computation, as we split the tokens into gradient and non-gradient segments for token-level control (see §3.2.2). While this adds minimal overhead, the trade-off is justified by the substantial memory savings and performance gains achieved by TO-KENSEEK. In sum, TOKENSEEK maintains training time efficiency comparable to other lightweight fine-tuning methods, while delivering superior performance and memory benefits.

Table S3: Training time for the main results in Tab. 1.

| | Baseline | Baseline + TOKENSEEK | LoHa | LoHa + TOKENSEEK | QLoRA | QLoRA + TOKENSEEK |
|---|---|---|---|---|---|---|
| **Qwen2.5** (0.5B) | | | | | | |
| GPU Hours | 0.43 | 0.49 | 0.62 | 0.69 | 0.77 | 0.86 |
| **Llama3.2** (1B) | | | | | | |
| GPU Hours | 0.35 | 0.39 | 0.73 | 0.82 | 0.55 | 0.63 |
| **Llama3.2** (3B) | | | | | | |
| GPU Hours | 0.67 | 0.75 | 1.63 | 1.85 | 1.03 | 1.18 |

## S1.7 DETAILS OF INSTRUCTION TUNING

All experiments are conducted on NVIDIA A100-40GB GPUs, except for the full token tuning settings on Llama3.2 3B due to the out-of-memory (OOM) issues. We train for one epoch using a learning rate of $4 \times 10^{-4}$ for all fine-tuning. A batch size of 1 is used with 32 gradient accumulation steps. Adapters are inserted into the feed-forward layers of each Transformer block following (He et al., 2021). The model is prompted using the Alpaca-style format (Taori et al., 2023) without explicit reasoning

## S2 DETAILS OF EFFICIENT TOKEN DITCHING

In this section, we present the implementation details of the efficient Token Ditching. Different from previous approaches, e.g., (Simoulin et al., 2024), TOKENSEEK introduces an instance-aware token selection framework, TOKENSEEK, which prioritizes tokens based on context and gradient-based information (see §3.2.1), thereby replacing random sampling with a principled, data-driven process. To facilitate understanding of the token ditching mechanism, we adopt the mathematical formulation below.

## S2.1 TOKEN DITCHING FOR DENSE AND NORMALIZATION LAYERS

For implementation, we adopt Algorithm 1 following (Simoulin et al., 2024), which explicitly partitions the hidden states into two subsets: $h_t$ for tokens selected for fine-tuning, and $h_{\bar{t}}$ for those gradient excluded. As illustrated in Eq.7 and Eq.8, the forward computation remains consistent with standard fine-tuning, with the key difference being that gradients are disabled for $h_{\bar{t}}$ using PyTorch's "torch.no_grad()" context, as shown in Eq. 8.

$$h_t = h_t W + b \tag{7}$$
$$h_{\bar{t}} = h_{\bar{t}} W + b \tag{8}$$

where $W$ represents the weights $W_1$ and $W_2$ of the feed-forward layers. A similar approach is applied to the normalization layers as well.

---

**Algorithm 1:** Token Ditching (To maintain clarity and focus, we simplify the model by removing layer normalization, skip connections, non-linear activations, and multi-head attention.)

---

**Input:** input tokens $X$
**Output:** selected tokens $h_t$, and unselected tokens $h_{\bar{t}}$

1   Compute token embeddings for the input sequence $h$

2   Divide the input tokens into two groups of selected and unselected tokens ($h_t$ and $h_{\bar{t}}$) via Token Seeking.

3   **for** $each\ transformers'\ layers$ **do**

     `// Attention computation`

4      $[Q_t, K_t, V_t] = h_t W_{[Q,K,V]} + b_{[Q,K,V]}$

5      $h_t = \mathrm{softmax}\left(\frac{Q_t[K_{\bar{t}}, K_t]^\top}{\sqrt{d}}\right)[V_{\bar{t}}, V_t]$

     `// No gradients for unselected tokens`

6      **with** torch.no_grad():

7         $[Q_{\bar{t}}, K_{\bar{t}}, V_{\bar{t}}] = h_{\bar{t}} W_{[Q,K,V]} + b_{[Q,K,V]}$

8         $h_{\bar{t}} = \mathrm{softmax}\left(\frac{Q_{\bar{t}}[K_{\bar{t}}, K_t]^\top}{\sqrt{d}}\right)[V_{\bar{t}}, V_t]$

     `// No gradients for unselected tokens`
     `// Feed-forward computation`

9      $h_t = h_t W_1 + b_1$

10     $h_t = h_t W_2 + b_2$

11     **with** torch.no_grad():

12        $h_{\bar{t}} = h_{\bar{t}} W_1 + b_1$

13        $h_{\bar{t}} = h_{\bar{t}} W_2 + b_2$

14   Re-organize input tokens into the original order

---

### S2.2   TOKEN DITCHING FOR ATTENTION LAYERS

For attention layers, we compute the attention as:

$$[Q_t, K_t, V_t] = h_t W_{[Q,K,V]} + b_{[Q,K,V]} \tag{9}$$

$$[Q_{\bar{t}}, K_{\bar{t}}, V_{\bar{t}}] = h_{\bar{t}} W_{[Q,K,V]} + b_{[Q,K,V]} \tag{10}$$

$$h_t = \mathrm{softmax}\left(Q_t[K_{\bar{t}}, K_t]^\top / \sqrt{d}\right)[V_{\bar{t}}, V_t] \tag{11}$$

$$h_{\bar{t}} = \mathrm{softmax}\left(Q_{\bar{t}}[K_{\bar{t}}, K_t]^\top / \sqrt{d}\right)[V_{\bar{t}}, V_t] \tag{12}$$

where $W_{[Q,K,V]} \in \mathbb{R}^{d \times 3d}$ represents the concatenated weights for the query, key, and value projections. For the unselected token positions in Eq.10 and Eq.12, gradient computation is again disabled using PyTorch. The complete forward pass procedure for the transformer model is detailed in Algorithm 1.

### S3   EXPERIMENTS ON LARGER LLMS

We further evaluate the potential of TOKENSEEK on larger models (7B scale) shown in Tab.S4. Due to resource constraints, only a subset of configurations is included in the results. As shown in the table, TOKENSEEK consistently outperforms TOKENTUNE across both full parameter/token tuning and QLoRA settings in terms of average score. Specifically, under full-token tuning, TOKENSEEK achieves an average score of 60.97, slightly improving over baseline (60.73), while reducing both average and peak memory by 22.1% and 54.4%, respectively. More notably, under the QLoRA setting, TOKENSEEK attains the comparable performance (62.14) while requiring only 18.2% average memory and 12.4% peak memory compared to full tuning, demonstrating a >80% memory saving. These results validate the scalability and robustness of our method even in large-model scenarios, reinforcing its value in memory-constrained training environments.

Table S4: Few-shot evaluation for Llama2 7B model. The experiments are performed under the same settings as Tab.1. † indicates the results reported from (Simoulin et al., 2024).

| Method | Ave. Mem. | Max. Mem. | ARC | Hella Swag | MMLU | Truthful QA | Wino Grande | Average Score |
|---|---|---|---|---|---|---|---|---|
| **Llama2** (7B) | | | | | | | | |
| Full Parameter/Token Tuning† | 100% | 100% | 52.39 | 78.97 | 64.44 | 38.97 | 68.90 | 60.73 |
| - w/ TOKENTUNE† | 77.9% | 45.6% | 51.71 | 78.35 | 61.56 | 41.88 | 70.01 | 60.70 |
| - w/ **TOKENSEEK** | **77.9%** | **45.6%** | 52.22 | 78.96 | 65.28 | 39.95 | 68.43 | **60.97** |
| QLoRA† | 53.2% | 46.5% | 56.06 | 78.60 | 65.08 | 43.64 | 69.38 | **62.55** |
| - w/ TOKENTUNE | 18.2% | 12.4% | 53.16 | 78.76 | 63.64 | 39.58 | 69.22 | 60.87 |
| - w/ **TOKENSEEK** | **18.2%** | **12.4%** | 53.50 | 78.82 | 65.26 | 44.62 | 68.51 | 62.14 |

## S4 ASSET LICENSE AND CONSENT

The majority of TOKENSEEK is released under the CC-BY-NC license. However, portions of the project are governed by separate license terms. Specifically, the Transformers library is licensed under Apache 2.0. Other dependencies used in this work include the HuggingFace PEFT and Datasets libraries, both under the Apache 2.0 license; the lm-evaluation-harness framework, which is licensed under MIT; and PyTorch, which is distributed under the modified BSD-3 license. The Open-Platypus dataset used for fine-tuning aggregates multiple datasets—detailed license information is available at `https://huggingface.co/datasets/garage-bAInd/Open-Platypus`.

## S5 REPRODUCIBILITY

Our implementation of TOKENSEEK is based on the HuggingFace Transformers library[1] (v4.33.1). For LoHa and QLoRA, we utilized the HuggingFace PEFT library[2] (v0.6.0). Datasets used for fine-tuning were obtained via the HuggingFace Datasets library[3] (v2.18.0), specifically using the Open-Platypus dataset[4].

For evaluation with the Qwen and Llama models, we employed the lm-evaluation-harness framework[5] (v0.4.2). All experiments were conducted using the PyTorch framework[6] (v2.0.1).

To guarantee reproducibility, our full implementation shall be publicly released upon paper acceptance.

## S6 SOCIAL IMPACT AND LIMITATIONS

TOKENSEEK presents a memory-efficient fine-tuning framework that significantly reduces training memory consumption while maintaining or even improving model performance. By selectively updating only the most informative tokens through an interpretable, instance-aware process, TOKENSEEK enables fine-tuning of LLMs on resource-constrained hardware. This advancement holds strong potential for democratizing LLM adaptation, making personalized and domain-specific model fine-tuning accessible in low-resource environments such as academic labs, startups, or edge devices. Moreover, TOKENSEEK aligns with broader goals of green AI by reducing computational and energy demands during training.

Despite these advantages, TOKENSEEK introduces a pre-stage evaluation that relies on two weighting factors ($\alpha$, $\beta$) to balance context and gradient-based token importance. While we empirically show in §4.4 that TOKENSEEK performs robustly across a wide range of settings, the selection of these hyperparameters introduces an additional tuning burden. In addition, TOKENSEEK requires a token evaluation step prior to training, which incurs additional computational overhead (*i.e.*, a forward

---

[1] `https://github.com/huggingface/transformers`
[2] `https://github.com/huggingface/peft`
[3] `https://github.com/huggingface/datasets`
[4] `https://huggingface.co/datasets/garage-bAInd/Open-Platypus`
[5] `https://github.com/EleutherAI/lm-evaluation-harness`
[6] `https://github.com/pytorch/pytorch`

pass and partial backward pass as discussed in §3.2.3). However, this trade-off is justified by the substantial memory savings and performance improvements achieved by TOKENSEEK. Furthermore, TOKENSEEK may be more sensitive to smaller-scale models, which possess limited representational capacity (Tab.1). These limitations suggest that a more lightweight and robust token evaluation could further improve the generality of our method.

In summary, TOKENSEEK contributes meaningfully toward the goal of efficient and scalable LLM adaptation, and we believe it offers valuable insights for future research in memory-efficient fine-tuning and token-level optimization.

## S7 DISCUSSION AND FUTURE WORK

### S7.1 DISCUSSION

**Regarding Token Selection Ratio Evaluation Gaps**. We have extended the study to cover both the low (<10%) and intermediate (50 – 100%) ranges under the Qwen2.5-0.5B QLoRA setting.

|  | 3% | 7% | 60% | 80% | 10% (comp.) |
|---|---|---|---|---|---|
| Average Score | 43.79 | 47.78 | 48.43 | 48.40 | **48.45** |
| Max. Mem. | 7% | 10% | 62% | 79% | 13% |

Table S5: Average score and maximum memory usage under different token evaluation ratios.

For low-ratio: Even under overly aggressive sparsity settings, TOKENSEEK maintains over 90% and 98% performance at the 3% and 7% settings, respectively, with only slight memory savings compared to the 10% setting. The observed performance degradation is likely due to the remaining tokens carrying insufficient gradient signals. For mid-to-high: Scores remain stable between 60% and 80%, indicating that TOKENSEEK continues to select high-quality tokens as the scale increases. However, memory usage rises sharply in this range, reducing efficiency. Overall, we recommend a 10% ratio as a balanced choice, considering both performance and memory efficiency.

**Regarding the Performance on Larger Scale Models**. We find that TOKENSEEK achieves more substantial performance gains when applied to smaller-scale models. We attribute this to a potential mismatch between model capacity and the training dataset size, since all models, regardless of scale, are fine-tuned on the same Open-Platypus dataset (25K samples), which may not fully exploit the capabilities of larger models. To investigate this, we conducted a preliminary experiment on the Llama2-7B model using an expanded dataset that adds 100K randomly sampled examples from MiniPile (Kaddour, 2023).

| Method | Dataset | Average Score |
|---|---|---|
| QLoRA | Open-Platypus (25K) | 62.55 |
| QLoRA + TOKENSEEK | Open-Platypus (25K) | 62.14 |
| QLoRA + TOKENSEEK | Open-Platypus (25K) + MiniPile (100K) | **63.26** |

Table S6: Average scores of QLoRA and QLoRA with TOKENSEEK on different datasets. Adding MiniPile (100K) to the training corpus improves performance.

As shown, incorporating more training samples yields additional performance improvements, supporting our hypothesis that the less favorable results on larger models may be due to the relatively small fine-tuning dataset.

**Regarding the Novelty and Differences**. Prior works assess token significance for token skipping in attention operations (Singh et al., 2024) or feature-importance explanations (Jain & Wallace, 2019). Conceptually different to these methods, TOKENSEEK evaluates tokens for gradient detaching, targeting memory savings.

TOKENSEEK leverages both context and gradient information, grounded in theoretical analysis and motivation. Initially, we use only context-based information to guide token selection, where we observe that higher scores tend to concentrate in earlier positions, an effect amplified by the causal mask. This bias may limit fine-tuning effectiveness, as the answer generation process is primarily

captured in later positions (*i.e.*, the "Response" portion of a training instance). This observation motivates the incorporation of gradient-based information to complement the context signals, enabling a more balanced and comprehensive evaluation of tokens. Together, this dual-perspective approach provides a flexible, plug-and-play solution for MEFT.

**Regarding Performance Gap in Llama Models**. Our implementations are based on Hugging Face PEFT, which provides a reliable and strong baseline for comparison. Regarding the performance gap observed in Llama but not in Qwen, this may stem from differences in how the two base models were built. Qwen-2.5-0.5B was trained at its target size from scratch (SFT $\rightarrow$ DPO / GRPO) (Yang et al., 2024), rather than being a pruned-and-distilled slice of a larger backbone. In contrast, Llama-3.2-1B/3B was created by first incorporating logits from the Llama-3.1-8B and 70B models as token-level targets. Knowledge distillation was then applied after pruning to recover performance (Grattafiori et al., 2024). This compression process results in sharper weights, which are therefore more sensitive (Bartoldson et al., 2020; Thangarasa et al., 2024) to gradient updates. As a result, full-parameter fine-tuning on a tiny dataset might drift off manifold, while PEFT methods that keep most weights frozen remain stable in Llama-3.2. These divergent construction pipelines account for the different behaviors observed in Table 1. We are very interested in this direction and plan to further investigate in the following work.

**Regarding the Regarding Task Diversity in Experimental Evaluation**. To further evaluate TOKENSEEK 's translation capability beyond code generation and mathematical reasoning, we use (Aharoni & Goldberg, 2020) as the training dataset and randomly sample 10K training examples from each domain (Medical, Law, IT, and Subtitles) to assess in-domain German-English translation performance under the Llama-2-7B. The preliminary BLEU scores are reported below, where we observe that TOKENSEEK consistently achieves comparable performance.

| Method | BLEU |
|---|---|
| Llama-2-7B | 33.13 |
| Llama-2-7B + LoRA | 40.16 |
| Llama-2-7B + LoRA + TOKENSEEK (10%) | **41.63** |

Table S7: BLEU scores of Llama-2-7B with LoRA and TOKENSEEK.

**More Baseline Comparison**. Due to page limitations in the main paper, we provide additional baseline comparisons here to offer a more complete view of memory usage and performance trends across methods.

| Metric | LoRA (1B) | IA3 (1B) | LoRA (3B) | IA3 (3B) | LoHa (7B) | IA3 (7B) |
|---|---|---|---|---|---|---|
| Max. Mem. | 92.6% | 88.9% | 90.1% | 85.2% | 90.5% | 84.2% |
| Average Score | 51.95 | 52.33 | 59.88 | 60.69 | 61.93 | 60.21 |

Table S8: Comparison of maximum memory usage and average scores across different parameter-efficient fine-tuning methods and model scales.

**Novelty Clarification with TOKENTUNE**. Although both methods aim to improve memory efficiency in fine-tuning, their underlying motivations, scoring mechanisms, and empirical behaviors differ fundamentally. TOKENTUNE relies on data-agnostic partial-gradient selection or random token dropping inspired by an engineering perspective, which we found to be ineffective and unstable across tasks. In contrast, TOKENSEEK is motivated by the observation that not all tokens contribute equally to model updates, and therefore adopts a data-driven, instance-aware criterion.

Specifically, TOKENSEEK introduces a hybrid scoring mechanism that integrates both attention-based contextual relevance and gradient-based optimization signals. This leads to substantially improved stability, interpretability, and effectiveness compared to random or data-agnostic selection. Beyond memory and performance metrics, TOKENSEEK also incorporates comprehensive analyses on stability, interpretability, and optimization behavior, offering insights into token-level contribution during efficient fine-tuning.

Finally, to the best of our knowledge, no prior MEFT method performs instance-aware token selection for activation-memory–efficient training. This instance-aware perspective represents the core novelty of TOKENSEEK and distinguishes it from TOKENTUNE's engineering-oriented design.

**The Claims of Memory Efficiency and Generality**. While TOKENSEEK adopts the same high-level token-ditching paradigm as TOKENTUNE, its advantages are substantially amplified due to our data-driven scoring design.

- **Memory reduction**. TOKENTUNE's random dropping causes unstable and degraded performance, forcing it to keep more tokens to stay competitive. In contrast, TOKENSEEK identifies truly salient tokens, enabling us to discard far more activations without hurting accuracy. As a result, under equal performance, TOKENSEEK consistently achieves significantly lower memory (*i.e.*, with only 10% tunable tokens, we achieve even higher performance than TOKENTUNE's 50% setting under the same model configuration, as shown in Fig. 3).
- **Generalizability**. TOKENSEEK relies solely on inherent signals from the pretrained model (attention and gradients). It requires no auxiliary model, no task-specific knowledge, and no architectural modification. This makes it compatible with any Transformer-based model and any PEFT method (Zeng et al., 2025b; Liu et al., 2025a). In contrast, several recent MEFT paradigms depend on customized modules (*e.g.*, reversible networks), which limits their applicability. TOKENSEEK remains universally plug-and-play across architectures and domains.

**Comparison with Sparsity-based PEFT**. Sparsity-based PEFT (Ansell et al., 2024; He et al., 2024; Frankle & Carbin, 2019) reduces memory use by updating only a small, selectively chosen subset of parameters instead of the full model during fine-tuning. In our main paper, we included BOFT as a representative sparsity-based PEFT method in Tab. 1. Here, we conducted additional experiments on RanLoRA (Albert et al., 2025) under the Qwen2.5 (0.5B) setting, as summarized below.

Table S9: Comparison of BOFT, RanLoRA, QLoRA, and QLoRA with TOKENSEEK under the Qwen2.5 0.5B setting.

| Method | Ave. Mem. | Max. Mem. | ARC | HellaSwag | MMLU | TruthfulQA | WinoGrande | Average Score |
|---|---|---|---|---|---|---|---|---|
| BOFT | 145.1% | 100.6% | 34.64 | 51.70 | 58.18 | 39.57 | 56.43 | 48.10 |
| RanLoRA | 95.4% | 86.7% | 29.18 | 50.10 | 58.33 | 45.21 | 57.22 | 48.01 |
| QLoRA | 51.7% | 45.6% | 34.64 | 50.10 | 58.05 | 40.41 | 55.09 | 47.66 |
| - w/ TOKENSEEK | 19.2% | 13.4% | 34.56 | 50.09 | 57.52 | 41.51 | 58.56 | **48.45** |

These results further enhance the comprehensiveness of our comparison and greatly deepen our paper demonstrate the effectiveness of our approach under the requested setting.

**Code-Domain Generalization**. We have conducted preliminary experiments under the Llama3.2 (1B) setting to evaluate the code-domain generalization as follows.

Table S10: Comparison of code-domain generalization under the Llama3.2 1B setting.

| Method | Ave. Mem. | Max. Mem. | ARC | HellaSwag | MMLU | TruthfulQA | WinoGrande | Humaneval | Average Score |
|---|---|---|---|---|---|---|---|---|---|
| **LoHa** | 92.3% | 99.4% | 39.25 | 65.93 | 57.60 | 37.87 | 60.77 | 13.41 | 45.81 |
| – w/ TOKENTUNE (Random) | 45.9% | 28.4% | 38.48 | 64.21 | 50.34 | 43.89 | 59.91 | 10.97 | 44.63 |
| – w/ TOKENSEEK (Ours) | 45.9% | 28.4% | 38.57 | 65.89 | 58.18 | 39.34 | 60.93 | 14.02 | 46.16 |
| **QLoRA** | 45.6% | 34.8% | 38.82 | 65.26 | 56.39 | 38.85 | 61.33 | 14.02 | 45.78 |
| – w/ TOKENTUNE (Random) | 14.8% | 14.3% | 39.33 | 62.97 | 41.76 | 41.36 | 60.69 | 12.80 | 43.15 |
| – w/ TOKENSEEK (Ours) | 14.8% | 14.3% | 39.08 | 65.98 | 58.03 | 38.65 | 61.33 | 14.63 | 46.28 |

Although coding tasks may contain denser information than QA tasks, TOKENSEEK still performs effectively, which may be because of our instance-aware token ditching and the strategy that we preserve the full forward pass, allowing complete attention and contextual information to remain intact.

**Discussion under the Distributed Environments**. We consider two major distributed fine-tuning settings: (1) DP: data-parallel training (including ZeRO/FSDP variants), and (2) TP/SP: tensor/sequence parallelism (model parallel training).

- For DP, each GPU holds a full copy of the model and only processes a different batch. Gradients are all-reduced at the end. DP has minimal impact on TOKENSEEK because every GPU handles its own local samples independently and only needs to synchronize parameters, not token-level information.

- For TP/SP, we would like to further analyze it separately since TOKENSEEK operates in two stages: instance-aware token seeking and efficient token ditching. The former is performed offline before training, while the latter is applied online during training.

  - For scoring, we involve the calculation of gradient score, which is computed only at the penultimate layer (all earlier layers are frozen), requiring substantially less memory than full fine-tuning (*i.e.*, requires only 13.2% memory of full fine tuning under qwen settings). It is inevitable to communicate gradients across GPUs to assemble the final gradient. However, because we compute gradients only in the penultimate layer, the additional memory cost remains manageable compared with full fine-tuning.

  - For training, we can distribute the token-score dictionary for each instance across GPUs before training, which consumes only minimal memory to store the mapping (*e.g.*, storing 50% of token positions for Open-Platypus requires about 6.8 MB). During training, this regrouping and reorganizing introduces an extra communication step for handling irregular gradient computation. However, thanks to efficient token ditching, the amount of gradient that needs to be synchronized is greatly reduced.

We are also looking forward to collaborating with extraordinary engineering teams to further optimize TOKENSEEK for more complex large-scale training scenarios.

**Reprouping Process under the Distributed Environments**. In conclusion, we provide a preliminary analysis of our unoptimized plain implementation and the communication challenges of applying TOKENSEEK in distributed environments. While these factors may reduce some of the memory savings observed in single-node training, the two-stage design combining partial-gradient scoring and partial-gradient updating keeps the overhead controllable compared with full fine-tuning.

In tensor parallelism, hidden dimensions are split across GPUs, so token regrouping is purely a local row reindexing operation. TP's usual all-reduce pattern stays unchanged. In sequence parallelism, however, the sequence dimension is sharded, so splitting tokens into selected and unselected sets breaks the local-contiguous token assumption. Each layer therefore requires an all-to-all shuffle to regroup tokens back to their original global order before proceeding. As a result, SP introduces small but necessary per-layer communication for token restoration.

**Complexity of Implementation**. Our implementation is based on huggingface's transformers and PEFT, which allows a single integration on one model to be directly reused and combined with other PEFT methods.

Specifically, we provide a detailed explanation of the modifications we make to each model below. We regroup the input I into [I_selected, I_unselected], apply "torch.no_grad()" to all I_unselected, and finally reorganize [O_selected, O_unselected] into the output O. This procedure is model-agnostic, follows a common pattern, and does not require manual adaptation to different model architectures, which can be handled by code agents that are highly capable of capturing these patterns, making the extension to other models straightforward.

**Selective Update Imbalance**. Dropping gradients for less important tokens may bias training if their importance is misestimated or varies across iterations. However, scoring and training designs enable TOKENSEEK to achieve stable evaluation (see mode details in the Section 4.3), align with the empirical results from Fig. 3.

We retain full-sequence attention and loss computation in the forward pass, and only zero out gradients for unselected tokens during backpropagation. This keeps the training objective and context intact while updating only the gradients deemed most important. This approach constitutes structured gradient sparsification rather than sample dropping or parameter pruning, and all parameters are still updated at every step, reducing the risk of systematic bias.

Beyond combining contextual and gradient-based signals to reduce potential misestimation from any single indicator, the scoring is derived from the current sample. It is therefore an instance-aware, dynamically updated selection mechanism rather than a fixed rule. Even if some iterations introduce noise, subsequent training iterations across many examples will adaptively mitigate it.

Furthermore, we also investigate the influences of misestimating token importance. Under the Llama3.2 (1B) QLoRA setup, we introduce an additional setting that selects Top 10% plus Top 40–50% tokens, instead of the standard Top 20%, to simulate misestimation.

Table S11: Ablation under misestimated token-importance settings for Llama3.2 1B QLoRA.

| Settings | Tunable Token | MMLU | ARC | HellaSwag | TruthfulQA | WinoGrande | Average Score |
|---|---|---|---|---|---|---|---|
| Random 20% | 20% | 40.10 | 63.93 | 42.96 | 43.23 | 61.01 | 50.25 |
| Top 20% | 20% | 39.42 | 65.73 | 53.24 | 39.86 | 60.77 | 51.80 |
| Top 10% + Top 40–50% | 20% | 39.16 | 65.91 | 51.20 | 39.28 | 61.01 | 51.31 |

Although "Top 10% + Top 40-50%" underperforms "Top 20%," it still outperforms "Random 20%," demonstrating the robustness of TOKENSEEK.

In the future, we plan to explore whether smoothing the scoring function or injecting a small portion of randomly selected tokens as exploration can further improve TOKENSEEK.

**Breakdown of Gradient Scoring**. We have added further quantification of our gradient scoring as summarized below.

Table S12: Gradient scoring breakdown for Qwen2.5 and Llama3.2 models.

| Model | Average Memory | Time (s) |
|---|---|---|
| Qwen2.5 (0.5B) | 13.2% | 291 |
| Llama3.2 (1B) | 11.5% | 377 |
| Llama3.2 (3B) | 10.9% | 566 |

Storing 50% of token positions for Open-Platypus requires about 6.8 MB.

We have also added the variance tables below.

Table S13: Variance of performance under different token evaluation ratios.

| 10% | 20% | 30% | 40% | 50% |
|---|---|---|---|---|
| 0.05242 | 0.16229 | 0.01396 | 0.00669 | 0.01620 |

**Automatic Learning of Hypeparameters**. Regarding the potential of learning $\alpha$ and $\beta$ automatically, grid search over {[1,0], [5,5], [7,3], [3,7]} on a validation set is generally practical and sufficient. In our case, Open-Platypus lacks a validation split, and our ablation in Tab. 2 shows that performance remains stable across these settings, indicating low sensitivity within this range and limited marginal benefit from learning them externally.

Regarding the potential of learning the token fraction $r$ automatically, we clarify that $r$ is a resource controller, which determines the number of tokens retained per sequence under a given memory budget. Our ablation in Fig. 6 shows that memory decreases sharply as $r$ moves from 50% down to 10%, making it more appropriate to choose $r$ based on the memory budget rather than learn it via a single objective such as validation loss. A budget-driven choice of $r$ better reflects the tradeoff between accuracy and memory savings.

**Gradient Score**. We conducted preliminary experiments under the Llama3.2 (1B) QLoRA setting using gradients from different layers as follows.

Table S14: Performance using gradients from different layers under the Llama3.2 (1B) QLoRA setting.

| Settings | MMLU | ARC | HellaSwag | TruthfulQA | WinoGrande | Average Score |
|---|---|---|---|---|---|---|
| Random | 39.33 | 62.97 | 41.76 | 41.36 | 60.69 | 49.22 |
| N-1 layer (default) | 39.08 | 65.98 | 58.03 | 38.65 | 61.33 | 52.61 |
| N-2 layer | 39.33 | 65.83 | 58.15 | 39.58 | 61.01 | 52.78 |
| N-3 layer | 39.08 | 65.60 | 57.55 | 39.27 | 61.17 | 52.53 |

From the results above, we do not observe obvious performance differences, and because using earlier layers requires storing more activation memory and introduce a new hyperparameter, the default setting is more practical.

Furthermore, we provide a deeper analysis of this pattern from a visualization perspective. We plot gradient scores obtained from the N-1 layer (blue), N-2 layer (orange), and N-3 layer (green), showing that the scoring pattern remains relatively stable across layers. It aligns with the empirical results above.

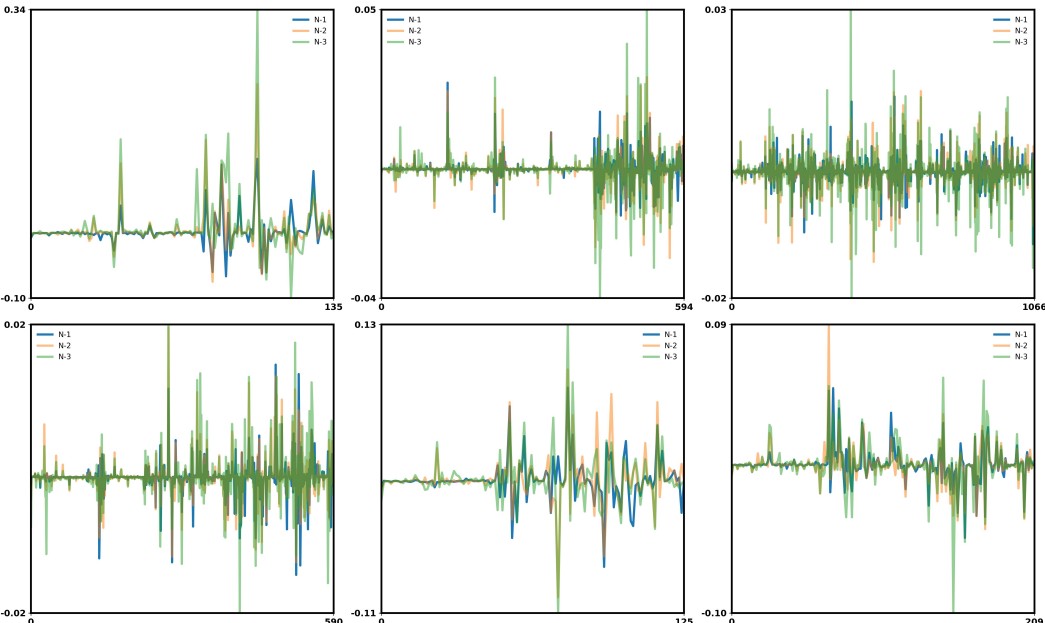

Figure S1: Additional visualization of gradient-based token scores across layers. The plot compares gradients from the N-1 (blue), N-2 (orange), and N-3 (green) layers.

**Attention Score**. The "global anchor", which is first introduced as attention sink phenomenon from [ref1], that refers to the tendency of attention to disproportionately concentrate on a single position, effectively acting as a global anchor. It serves as a reference token (*i.e.*, the baseline) for scoring all subsequent tokens.

In TOKENSEEK, the same attention sink effect emerges (see Fig.4 (b)) because the model naturally assigns one position an abnormally large and stable amount of attention. This token becomes the model's internal reference point. Since TOKENSEEK scores tokens using both forward attention signals and backward gradient signals, this "sink" position nearly always receives a high combined score even after the normalization (see Fig.4 (d)). As a result, it is consistently preserved rather than ditched. TOKENSEEK does not artificially enforce this behavior. It simply reflects the model's inherent dynamics, where the sink token provides a stable baseline that anchors attention patterns across layers and steps.

**Token Efficiency across Different Domains**. A similar token efficiency phenomenon has been observed in LLM reinforcement learning (RL) and supervised fine-tuning (SFT) reasoning research (Wang et al., 2025; Qian et al., 2025).

Both domains converge on the observation that model behavior is disproportionately shaped by a small subset of influential tokens. In the reasoning literature, this is reflected in localized spikes in mutual information or high-entropy branching positions that largely determine the downstream reasoning trajectory. TOKENSEEK arrives at a parallel conclusion from the training perspective: by jointly examining gradient magnitude and attention allocation, we find that only a minority of tokens make substantial stable contributions to parameter updates and thus to effective fine-tuning. In both cases, the model does not treat all tokens equally. Instead, it implicitly concentrates its computational and learning capacity on structurally or semantically pivotal positions.

Despite this shared principle, the operational signals, and scopes of the two lines of work are fundamentally distinct. 1) Signal Source: reasoning work relies on inference-time indicators such as mutual information spikes or entropy changes, while TOKENSEEK combines forward attention and backward gradients to estimate how much each token contributes to loss reduction, and then selectively allocates training budget accordingly. 2) Scope: reasoning research typically highlights a handful of discrete "turning point" tokens, whereas TOKENSEEK evaluates the entire sequence to identify all tokens that meaningfully influence parameter updates.

Consequently, although both areas reveal token-level sparsity in model computation, they capture different facets of model behavior and operate under different optimization goals. In the future, we would like to further explore how our token scoring strategy might be extended to reasoning.

**Relation to RL-based Reasoning and Token-level Analyses**. Recent work (Wang et al., 2025; Qian et al., 2025; Lin et al., 2024; Zeng et al., 2025c; Luo et al., 2026; Zhou et al., 2025; Lightman et al., 2023; Yao et al., 2025; Guo et al., 2025; Ren et al., 2025) on RL-based reasoning and chain-of-thought analyses consistently shows that only a small fraction of tokens carry most of the useful learning or information signal. RLVR (Wang et al., 2025) finds that high-entropy "forking" tokens account for nearly all performance gains in mathematical reasoning, while gradients on low-entropy tokens contribute little or even harm accuracy. Similarly, mutual-information analyses (Qian et al., 2025) identify sparse "MI peaks" whose "thinking tokens" are crucial for final-answer prediction, suppressing these tokens severely degrades reasoning. These studies collectively provide a fine-grained view of where RL-style updates and inference-time computation actually matter.

DeepSeekMath (Shao et al., 2024) further links SFT and RL by showing that RL methods like GRPO can be interpreted as reshaping gradients while staying close to a supervised reference model. In this view, both TOKENSEEK and RLVR adopt token-level importance as the core abstraction, but operate at different stages and with different signals. RLVR prioritizes high-entropy tokens during policy updates while TOKENSEEK reallocates the SFT gradient budget across tokens and prioritizes high-score tokens during SFT. In this sense, TOKENSEEK addresses the problem of token-wise efficiency from a complementary angle with current reasoning work: we focus on memory-efficient gradient allocation in supervised fine-tuning, while prior RL-based reasoning studies focus on token-wise credit assignment and information flow during policy optimization and inference.

These connections motivate us a future work extension in the spirit of (Wang et al., 2025; Zeng et al., 2025a; Qian et al., 2025). TOKENSEEK scoring function is deliberately restricted to signals available in a standard SFT pipeline, but it would be natural in future work to augment TOKENSEEK with additional token-level diagnostics (*e.g.*, entropy or MI estimates) or to design hybrid schedules where SFT and RL share a common token-importance backbone.

**Ethics Statement**. We conform to the ICLR Code of Ethics and further show the consent to our work below. All datasets used in this study are publicly available and released under permissive licenses (see §S4), and all the models are publicly available (see §S4 for Asset License and Consent). We would like to state that the contents in the dataset do NOT represent our views or opinions and our paper does not involve crowdsourcing or research with human subjects.

**AI Disclosure**. We acknowledge the use of GPT-5 for grammar correction and sentence-level refinement only. The model was employed to enhance clarity, coherence, and fluency while ensuring the original meaning and intent of the text remained unchanged.

## S7.2 FUTURE WORK

In §2, we review existing PEFT and MEFT methods, highlighting their focus on optimizing different components of the training pipeline. Unlike prior data-agnostic approaches, TOKENSEEK introduces an instance-aware mechanism that combines context and gradient information to identify and retain the most informative tokens during fine-tuning. Despite the effectiveness and generality of TOKENSEEK, it raises several open questions and directions for future research. One current limitation lies in the manual selection of weighting scalars $\alpha$ and $\beta$, which control the influence of context and gradient signals. While we provide empirical guidance on effective ranges (see §4.4), developing an automated mechanism—such as a lightweight controller or hypernetwork—to learn these weights adaptively could enhance performance and reduce tuning overhead.

Another promising direction lies in extending TOKENSEEK beyond instruction tuning and classification tasks to more complex settings such as multi-modal fine-tuning or continual learning. Additionally, although TOKENSEEK integrates well with PEFT methods like LoHa and QLoRA (see Tab.1), further exploration is needed to evaluate its synergy with sparse or retrieval-augmented architectures.

Lastly, while TOKENSEEK demonstrates strong interpretability and robustness across multiple LLMs, deeper analysis of its token evaluation patterns across domains (*e.g.*, code, biomedical texts) may offer insights into task-specific redundancy and inform domain-adaptive pruning strategies.

In summary, TOKENSEEK presents a general and interpretable framework for memory-efficient fine-tuning. Future work can build on this foundation by exploring automated token selection, broader task applicability, and tighter integration with emerging efficient model designs.

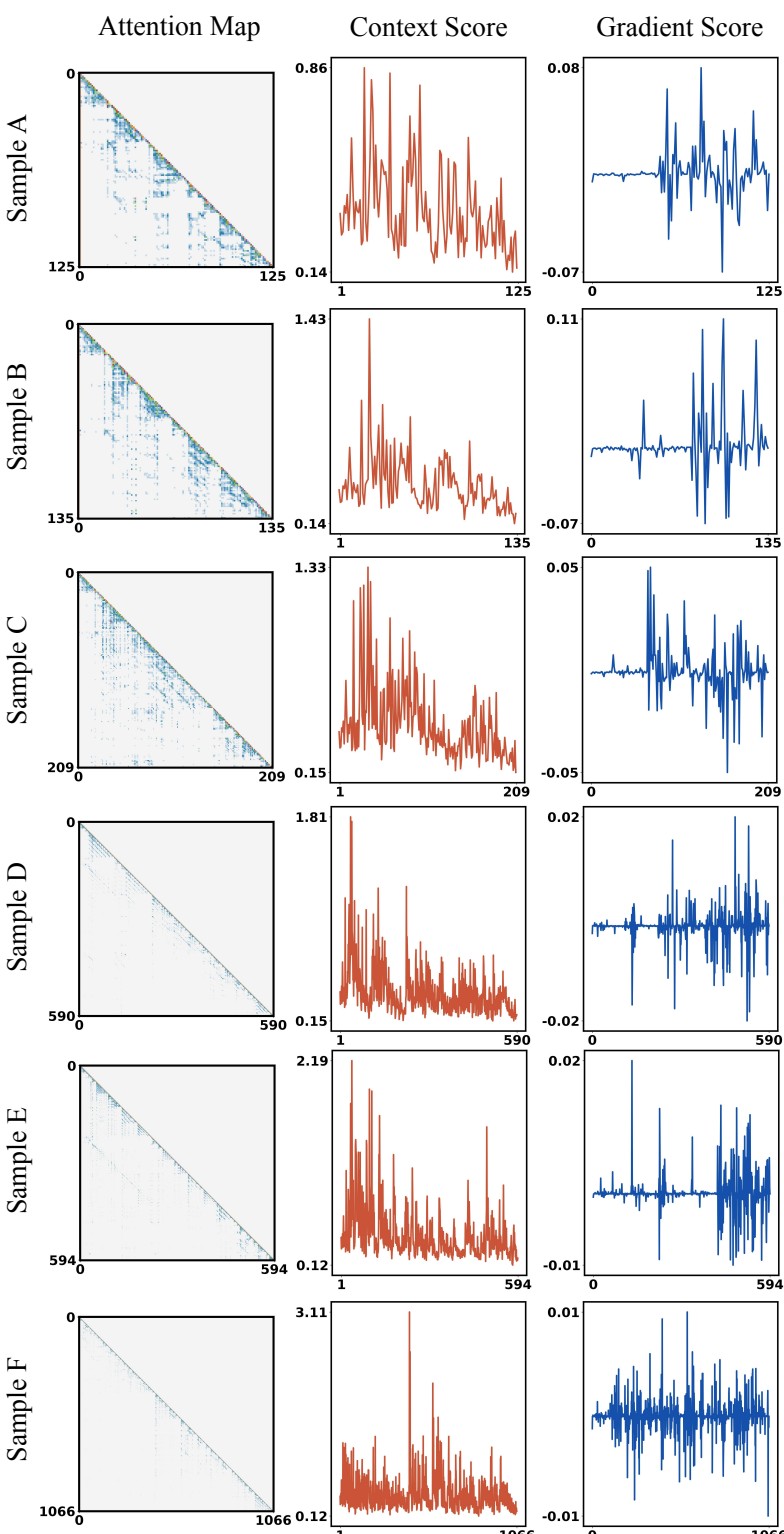

Figure S2: Additional visualizations of the attention map, context-based scores, and gradient-based scores. For better observation, we omit the attention score of the first token (*i.e.*, context scores start from position 1 instead of 0) due to the attention sink phenomenon discussed in §4.3.

## S8   VISUALIZATION OF ATTENTION AND GRADIENT MAPS

Fig.S2 presents attention maps, context-based token scores, and gradient-based scores across six randomly selected samples, ranging in length from 126 to 1067 tokens. These examples provide further evidence supporting the key findings from the main paper in §4.3. Across all samples, we observe a consistent pattern in context-based scores, where higher values are concentrated in the earlier token positions (*i.e.*, a manifestation of the commonly observed attention sink effect and causal masking in autoregressive models). This phenomenon causes tokens in initial positions to accumulate more attention, aligning with prior observations discussed in §4.3. In contrast, gradient-based token scores tend to emphasize later positions, particularly toward the response segments in instruction tuning tasks. This reflects the model's training dynamics: gradients are more pronounced where output predictions are made and optimized—typically in the latter portion of the sequence. Despite differences in sequence lengths, this divergence in focus between the two signals remains stable across all samples. These findings reinforce the motivation behind combining both context and gradient signals in TOKENSEEK for a more comprehensive and balanced token importance evaluation.

---

Tokens selected by context score

**Below** is an **instruction that describes** a **task. Write** a **response that appropriately completes** the request.

### Instruction**:**
**Evaluate $(x+ y)(x-y)$ when $x= 15$** and $y= **5**$.

### Response:
**To evaluate the expression, I** need **to plug in** the given **values** of $x$ and $y$ and **simplify using the** order of operations. So I **get $(15**+ **5**)(**15**-5)$. **I can do the addition** and subtraction **inside** the parentheses first, and get **$(20)**(10)$. Then I **multiply** 20 and 10 and get 200.

- - - - - - - - - - - - - - - - - - - - - - - - - - - - - - - - - - - - - - - - - -

Tokens selected by gradient score

**Below is** an instruction that **describes** a task. **Write** a response that **appropriately completes** the request.

### Instruction:
Evaluate $(x+ y**)(x-y)$ when $x**= 15$** and **$y= 5**$.

### **Response:**
To **evaluate** the **expression,** I need to **plug in the** given **values of** $x$ **and $y$** and **simplify** using **the** order of **operations. So I** get **$(15**+ **5**)(15-5)$. I **can do the** addition and subtraction inside **the parentheses** first, and get $(20**)(10)$. Then **I multiply 20** and **10 and get 200**.

---

Figure S3: Visualization of the top 50% selected tokens in Sample A presented in Fig.S2 using context and gradient information, highlighted in red and blue, respectively.

Fig.S3 presents a case study of a 126-token instruction-response Sample A presented in Fig.S2, with the top 50% tokens highlighted based on importance scores derived from context and gradient information, respectively. Tokens highlighted in red correspond to the top 50% according to context importance, while those in blue correspond to the top 50% based on gradient-based importance.

From the Fig.S3, we observe a clear distributional pattern in the highlighted tokens: **Context-based selection** (red) tends to emphasize tokens in the earlier part of the sequence—particularly the instruction prompt and structural phrases such as "Below $\cdots$ instruction that describes $\cdots$ task" and "Evaluate (x+y)(x-y) when $\cdots$". In contrast, **gradient-based selection** (blue) focuses more on the response portion, especially on semantically meaningful action words such as "evaluate", "plug in", "simplify", and numerical reasoning steps like "multiply 20 $\cdots$ 10 and get 200". These tokens are closely tied to the model's output prediction and loss computation, which naturally generate higher gradients. Interestingly, while there is some overlap (*e.g.*, tokens like "$x$", "$y$", and the equation), the two selection strategies yield complementary subsets, justifying the motivation for combining them in TOKENSEEK. This example visually supports the core hypothesis that instance-aware token prioritization benefits from incorporating both structural (context) and optimization-relevant (gradient) information.

Fig.S4 presents a long-form, math-intensive instruction-response pair consisting of 1067 tokens, where the top 50% of tokens are selected solely based on gradient-based importance. This sample offers several interesting insights into the behavior of TOKENSEEK 's gradient-driven token prioritization mechanism. There are three Key Observations:

- **Gradient Emphasis on Semantically Dense Mathematical Reasoning**. Tokens receiving high gradient scores are concentrated around numerical reasoning, symbolic manipulation, and step-by-step algebraic deduction. For example: ❶ The derivation and solving of equations such as "$x^2 + y^2 = 16.$". ❷ Geometry-specific calculations like "$\frac{24\sqrt{2}}{5} \cdot \frac{8\sqrt{7}}{5}$", and ❸ Descriptions of intersections, area computation, and final results. This confirms that gradient-based importance scores effectively highlight regions where model predictions are tightly coupled with loss, especially in problem-solving and logic-intensive portions of the response.

- **Selective Attention in Code Blocks** Surprisingly, some code comments and critical semantic structures in the embedded code block are also assigned high gradient scores. This includes: ❶ Function calls like draw(...), label(...), and array manipulations involving coordinates. ❷ Mathematical graphing logic, such as drawing a circle or intersecting paths with graph(...). This suggests that gradient signals are not purely confined to natural language but can also prioritize symbolic logic and programmatic structures that are critical to the correct final output.

- **Omission of Setup and Template Tokens**. The gradient-based selection intentionally avoids early template phrases (*e.g.*, "· · · instruction · · · describes · · · task") and instead defers attention to content-bearing tokens, particularly in the response body. This is aligned with prior gradient analysis (see §3.2.1), and reinforces the idea that gradient prioritization aligns well with actual supervision information.

Below is an **instruction** that **describes** a **task**. Write a **response** that **appropriately completes the request**.

### Instruction:
Let $\mathcal{H}$ be **the hyper**bola **with** foci at $(\pm 5, 0)$ **and** vertices at $(\pm 3, 0),$ **and** let $\mathcal{C}$ **be the** circle with **center** $(0,0)$ and **radius** $4.$ Given that $\mathcal{H}$ and $\mathcal{C}$ **intersect** at **four points,** what **is the** area **of the quadrilateral formed by the four** points?

### **Response:**
**For** the **hyper**bola $\mathcal{H},$ **we** have $a=3$ and $c=5,$ so $b=\sqrt{c^2-a^2} = 4.$ **Thus**, the **hyperbola has equation** \[\frac{x^2}{3^2} - _[\frac{y^2}{4^2} = 1,\]**or** \[16x^2 - 9y^2 = 144.\]**Meanwhile**, the equation **for the** circle is $x^2 + y^2 = 16.$ To **find** the **points of** intersection, we solve **these** two **equations** simultaneously. Adding $9$ **times** the second equation **to the** first **equation** gives $25x^2 = 288,$ so $x = \pm \frac{12\sqrt2}{5}.$ **Then** we have \[y^2 = 16 - x^2 = 16 - \frac{288}{25} = \frac{112}{25},\]so $y = \pm \frac{4\sqrt7}{5}.$ **Therefore**, **the** four **points of intersection** form **a rectangle** with side lengths $\frac{24\sqrt2}{5}$ **and** $\frac{8\sqrt7}{5},$ **so** its **area** is $\frac{24\sqrt2}{5} \cdot \frac{8\sqrt7}{5} = \frac{192\sqrt{14}}{25}.$

```
[asy]
void axes(real x0, real x1, real y0, real y1)
{
    draw((x0,0)--(x1,0),EndArrow);
    draw((0,y0)--(0,y1),EndArrow);
    label("$x$",(x1,0),E);
    label("$y$",(0,y1),N);
    for (int i=floor(x0)+1; i<x1; ++i)
        draw((i,.1)--(i,-.1));
    for (int i=floor(y0)+1; i<y1; ++i)
        draw((.1,i)--(-.1,i));
}
path[] yh(real a, real b, real h, real k, real x0, real x1, bool upper=true, bool lower=true, pen color=black)
{
    real f(real x) { return k + a / b * sqrt(b^2 + (x-h)^2); }
    real g(real x) { return k - a / b * sqrt(b^2 + (x-h)^2); }
    if (upper) { draw(graph(f, x0, x1),color, Arrows); }
    if (lower) { draw(graph(g, x0, x1),color, Arrows); }
    path [] arr = {graph(f, x0, x1), graph(g, x0, x1)};
    return arr;
}
void xh(real a, real b, real h, real k, real y0, real y1, bool right=true, bool left=true, pen color=black)
{
    path [] arr = yh(a, b, k, h, y0, y1, false, false);
    if (right) draw(reflect((0,0),(1,1))*arr[0],color, Arrows);
    if (left) draw(reflect((0,0),(1,1))*arr[1],color, Arrows);
}
void e(real a, real b, real h, real k)
{
    draw(shift((h,k))*scale(a,b)*unitcircle);
}
size(8cm);
axes(-6,6,-6,6);
xh(3,4,0,0,-5,5);
e(4,4,0,0);
dot((5,0)^^(-5,0)^^(3,0)^^(-3,0));
for (int i=-1; i<=1; i+=2)
    for (int j=-1; j<=1; j+=2)
    dot((i*12*sqrt(2)/5,j*4*sqrt(7)/5));
draw((-1*12*sqrt(2)/5,-1*4*sqrt(7)/5)--(12*sqrt(2)/5,-1*4*sqrt(7)/5)--(12*sqrt(2)/5,4*sqrt(7)/5)--(-
12*sqrt(2)/5,4*sqrt(7)/5)--cycle,dotted);
[/asy]
```

Figure S4: Visualization of the top 50% selected tokens in Sample F presented in Fig.S2 using gradient information, highlighted in blue.

