# OpenReview forum: "TokenSeek: Memory Efficient Fine Tuning via Instance-Aware Token Ditching"
_ICLR.cc/2026/Conference — ICLR 2026 Poster_

### Official Review · Reviewer_cH6y · 2025-10-20

**Soundness:** 2
**Presentation:** 3
**Contribution:** 2
**Rating:** 4
**Confidence:** 4

**Summary:**

This paper proposes TOKENSEEK, a method for memory-efficient fine-tuning (MEFT) of LLMs that addresses the bottleneck of activation memory. TOKENSEEK introduces an instance-aware paradigm that dynamically identifies and preserves the most informative tokens per input instance while discarding less crucial ones during gradient computation.

**Strengths:**

>s1: The key insight behind TOKENSEEK is that not all training tokens within LLMs contribute equally to
model fine-tuning.

>s2: The token selection process is based on a transparent, hybrid scoring mechanism leveraging both contextual attention and gradient information. This provides interpretability into why certain tokens are selected and contributes to more stable and effective fine-tuning compared to data-agnostic or random baselines.

>s3: The presentation is clear, and the figures are of high quality.

**Weaknesses:**

See Questions. I would reconsider my score if these concerns are adequately addressed.

**Questions:**

> w1: Regarding the computation of gradient information, the description in Lines 230-232 specifies the use of "the activations in the penultimate layer," and Lines 262-263 mention freezing "all layers except the output head and the final decoder block." Was the use of gradients from more layers explored? Furthermore, would expanding the scope of gradient computation in this manner still ensure stable fine-tuning, or could it potentially introduce instability?

> w2: The manuscript introduces scalars α and β to integrate context and gradient information (Lines 236-245) and notes their "distinct but complementary patterns" (Lines 405-407). However, the rationale and process for selecting the optimal values for α and β are not sufficiently discussed. A more detailed analysis is needed to clarify how these critical hyperparameters were determined and how sensitive the method's performance is to their specific values.

>w3: The authors identify a "global anchor" in the attention map (Lines 373-377), evident as a prominent vertical line. Based on the proposed TokenSeek framework, could the authors provide an explanation for why this specific token emerges as a global anchor? Furthermore, given that your method ditches tokens based on a combined score, is this particular "anchor" token ever identified as less informative and subsequently ditched, or is it consistently preserved? Clarifying this would enhance the interpretability of the token selection process.

>w4: The insight that not all training tokens contribute equally to model fine-tuning is well-founded and reasonable. A similar phenomenon has been observed in LLM reasoning research [1,2]. Could the authors analyze the connections and distinctions between your findings and those from the reasoning domain?
>
>[1] Beyond the 80/20 Rule: High-Entropy Minority Tokens Drive Effective Reinforcement Learning for LLM Reasoning.
>
>[2] Demystifying Reasoning Dynamics with Mutual Information: Thinking Tokens are Information Peaks in LLM Reasoning.

---

> ### Author Response · Authors · 2025-11-19
> **Author Response to Reviewer cH6y (Part 1/2)**
>
> Thank you for acknowledging the methodological clarity, interpretability, and empirical rigor of our method. We sincerely appreciate your time and effort in reviewing our paper and providing valuable comments. We provide explanations to your questions point-by-point in the following.
>
> ---
> ### **${\color{#f26921}\text{R1: Regarding the Computation of Gradient Information.}}$**
>
> > ${\color{#f26921}\text{Question-1:}}$ "Regarding the computation of gradient information, the description in Lines 230-232 specifies the use of "the activations in the penultimate layer," and Lines 262-263 mention freezing "all layers except the output head and the final decoder block." Was the use of gradients from more layers explored? Furthermore, would expanding the scope of gradient computation in this manner still ensure stable fine-tuning, or could it potentially introduce instability?."
>
> ${\color{#f26921}\text{Response Q1:}}$
>
> Thank you for this insightful suggestion. Following your comments, we conducted preliminary experiments under the Llama3.2 (1B) QLoRA setting using gradients from different layers as follows.
>
> | Settings            | MMLU  | ARC   | Hella Swag | Truthful QA | Wino Grande | Average Score |
> | ------------------- | ----- | ----- | ---------- | ----------- | ----------- | ------------- |
> | Random              | 39.33 | 62.97 | 41.76      | 41.36       | 60.69       | 49.22         |
> | N-1 layer (default) | 39.08 | 65.98 | 58.03      | 38.65       | 61.33       | 52.61         |
> | N-2 layer           | 39.33 | 65.83 | 58.15      | 39.58       | 61.01       | 52.78         |
> | N-3 layer           | 39.08 | 65.60 | 57.55      | 39.27       | 61.17       | 52.53         |
>
>
> From the results above, we do not observe obvious performance differences, and because using earlier layers requires storing more activation memory and introduce a new hyperparameter, the default setting is more practical.
>
> Furthermore, we provide a deeper analysis of this pattern from a visualization perspective. We plot gradient scores obtained from the N-1 layer (blue), N-2 layer (orange), and N-3 layer (green) in Page 25, showing that the scoring pattern remains relatively stable across layers. It aligns with the empirical results above.
>
> ---
> ### **${\color{#f26921}\text{R2: Regarding the Hyperparameters.}}$**
>
> > ${\color{#f26921}\text{Question-2:}}$ "The manuscript introduces scalars α and β to integrate context and gradient information (Lines 236-245) and notes their "distinct but complementary patterns" (Lines 405-407). However, the rationale and process for selecting the optimal values for α and β are not sufficiently discussed. A more detailed analysis is needed to clarify how these critical hyperparameters were determined and how sensitive the method's performance is to their specific values."
>
> ${\color{#f26921}\text{Response Q2:}}$
>
> Thank you for this excellent question. We fully agree with the reviewer that a detailed analysis of these hyperparameters is important to our paper, and we provide a clearer clarification here.
>
> As discussed in Appendix S1.3, we perform a linear search over the set {\[1,0\], \[0,1\], \[5,5\]} for each LLM. It reveals that smaller models (Qwen2.5 0.5B, Llama3.2 1B) benefit from context-only information, while the larger Llama3.2 3B performs the best when combining both context and gradient information. This suggests that larger models may better utilize multi-information token evaluation.
>
> Regarding the sensitivity of these hyperparameters, our ablation in Table 2 shows that performance remains stable across {\[1,0\], \[5,5\], \[7,3\], \[3,7\]} settings, indicating low sensitivity within certain range.

---

> ### Author Response · Authors · 2025-11-19
> **Author Response to Reviewer cH6y (Part 2/2)**
>
> ### **${\color{#f26921}\text{R3: Regarding the Global Anchor.}}$**
>
> > ${\color{#f26921}\text{Question-3:}}$ "The authors identify a "global anchor" in the attention map (Lines 373-377), evident as a prominent vertical line. Based on the proposed TokenSeek framework, could the authors provide an explanation for why this specific token emerges as a global anchor? Furthermore, given that your method ditches tokens based on a combined score, is this particular "anchor" token ever identified as less informative and subsequently ditched, or is it consistently preserved? Clarifying this would enhance the interpretability of the token selection process."
>
> ${\color{#f26921}\text{Response Q3:}}$
>
> Thank you for the insightful observation regarding the “global anchor”, which is an important and consistent phenomenon that aligns with our empirical findings. A more detailed explanation is provided below.
>
> The “global anchor”, which is first introduced as attention sink phenomenon from \[ref1\], that refers to the tendency of attention to disproportionately concentrate on a single position, effectively acting as a global anchor. It serves as a reference token (i.e., the baseline) for scoring all subsequent tokens.
>
> In TokenSeek, the same attention sink effect emerges (see Figure 4 (b)) because the model naturally assigns one position an abnormally large and stable amount of attention. This token becomes the model’s internal reference point. Since TokenSeek scores tokens using both forward attention signals and backward gradient signals, this “sink” position nearly always receives a high combined score even after the normalization (see Figure 4 (d)). As a result, it is **consistently preserved** rather than ditched. TokenSeek does not artificially enforce this behavior. It simply reflects the model’s inherent dynamics, where the sink token provides a stable baseline that anchors attention patterns across layers and steps.
>
> We have supplemented the discussion into our revision.
>
> ---
> ### **${\color{#f26921}\text{R4: Regarding the Further Discussion.}}$**
>
> > ${\color{#f26921}\text{Question-4:}}$ "The insight that not all training tokens contribute equally to model fine-tuning is well-founded and reasonable. A similar phenomenon has been observed in LLM reasoning research \[1,2\]. Could the authors analyze the connections and distinctions between your findings and those from the reasoning domain?"
>
> ${\color{#f26921}\text{Response Q4:}}$
>
> Thank you for the excellent suggestion, which greatly improve the quality of our work. We would love to discuss token efficiency across different domains.
>
> Commonality:
> Both domains converge on the observation that model behavior is disproportionately shaped by a small subset of influential tokens. In the reasoning literature, this is reflected in localized spikes in mutual information or high-entropy branching positions that largely determine the downstream reasoning trajectory. TokenSeek arrives at a parallel conclusion from the training perspective: by jointly examining gradient magnitude and attention allocation, we find that only a minority of tokens make substantial stable contributions to parameter updates and thus to effective fine-tuning. In both cases, the model does not treat all tokens equally. Instead, it implicitly concentrates its computational and learning capacity on structurally or semantically pivotal positions.
>
> Differences:
> Despite this shared principle, the objectives, operational signals, and scopes of the two lines of work are fundamentally distinct. **1\) Objective:** reasoning studies aim to understand how models carry out multi-step inference, whereas TokenSeek focuses on memory-efficient training by identifying which tokens meaningfully drive parameter updates. **2\) Signal Source:** reasoning work relies on inference-time indicators such as mutual information spikes or entropy changes, while TokenSeek combines forward attention and backward gradients to estimate how much each token contributes to loss reduction, and then selectively allocates training budget accordingly. **3\) Scope:** reasoning research typically highlights a handful of discrete “turning point” tokens, whereas TokenSeek evaluates the entire sequence to identify all tokens that meaningfully influence parameter updates.
>
> Consequently, although both areas reveal token-level sparsity in model computation, they capture different facets of model behavior and operate under different optimization goals.
>
> We have incorporated these citations and discussion into our revision. In the future, we would like to further explore how our token scoring strategy might be extended to reasoning.
>
> ---
>
> All modifications have been marked in ${\\color{\#f26921} orange}$ in our revised submission.
>
> **We appreciate your thoughtful comments. We hope our response addresses your concerns. Please let us know if there are any additional questions, and we will be happy to discuss further.**

---

> > ### Comment · Reviewer_cH6y · 2025-11-26
> > **Thanks for your response, still have some concerns about W4.**
> >
> > Thank you for the extensive and thoughtful rebuttal. Many of my earlier concerns have been addressed, and I appreciate the authors’ effort in improving the clarity and organization of the paper.
> >
> > However, the issue raised in W4 still feels insufficiently resolved.
> >
> > For your response：
> >
> > > Despite this shared principle, the objectives, operational signals, and scopes of the two lines of work are fundamentally distinct. 1) Objective: reasoning studies aim to understand how models carry out multi-step inference, whereas TokenSeek focuses on memory-efficient training by identifying which tokens meaningfully drive parameter updates. 2) Signal Source: reasoning work relies on inference-time indicators such as mutual information spikes or entropy changes, while TokenSeek combines forward attention and backward gradients to estimate how much each token contributes to loss reduction, and then selectively allocates training budget accordingly. 3) Scope: reasoning research typically highlights a handful of discrete “turning point” tokens, whereas TokenSeek evaluates the entire sequence to identify all tokens that meaningfully influence parameter updates.
> >
> > I do not believe that other related works (such as [1]) rely on this Objective as an excuse; for example, [1] also identify which tokens meaningfully drive parameter updates——"In contrast, training exclusively on the 80% lowest-entropy tokens leads to a marked decline in performance".
> >
> > Given the short rebuttal window, I understand it may be difficult to fully resolve. I encourage the authors to consider addressing this point in a future revision, either by:
> > (1) Extending the analysis between RL and SFT like [2], or (2) Complementing the token level discussion.
> >
> > [1] Beyond the 80/20 Rule: High-Entropy Minority Tokens Drive Effective Reinforcement Learning for LLM Reasoning.
> >
> > [2] DeepSeekMath: Pushing the Limits of Mathematical Reasoning in Open Language Models.
> >
> > That said, taking other improvements into account, I am inclined to raise my score from 4 to 6.

---

> ### Author Response · Authors · 2025-11-27
> **Thank you for engaging the discussion!**
>
> Thank you for your positive feedback, your recognition of our work, and your willingness to adjust the rating. We are glad that your major concerns have been addressed through our additional discussion and results.
>
> We fully agree that prior RL-based works also identify which tokens meaningfully drive parameter updates, and we have revised the corresponding phrasing in our paper.
>
> Following your suggestion, we added a preliminary discussion, given the short rebuttal window,  in Appendix S7.1 (page 26) connecting our method to RL-based reasoning and token-level analyses. We completely understand your concerns about W4. **We commit to thoroughly and continually** incorporating these discussions and comparisons [ref1-11] into the revision, ensuring a deeper and more comprehensive analysis of token importance.
>
> Furthermore, we very much appreciate the reviewer’s suggestions, as they illuminate a clear direction for future work on exploring hybrid training schedules in which SFT and RL token credit assignment may share a common token-evaluation backbone.
>
> We truly appreciate your thoughtful consideration and **constructive engagement** with our rebuttal - it has significantly strengthened our work.
>
> ---
>
> **References:**
>
> [1] Beyond the 80/20 Rule: High-Entropy Minority Tokens Drive Effective Reinforcement Learning for LLM Reasoning. NeurIPS 2025.
>
> [2] Demystifying Reasoning Dynamics with Mutual Information: Thinking Tokens are Information Peaks in LLM Reasoning. Arxiv 2025.
>
> [3] DeepSeekMath: Pushing the Limits of Mathematical Reasoning in Open Language Models. Arxiv 2024.
>
> [4] Critical Tokens Matter: Token-Level Contrastive Estimation Enhances LLM's Reasoning Capability. ICML 2025.
>
> [5] Token-level Direct Preference Optimization. ICML 2024.
>
> [6] T-REG: Preference Optimization with Token-Level Reward Regularization. ACL 2025.
>
> [7] Does reinforcement learning really incentivize reasoning capacity in llms beyond the base model? Arxiv 2025.
>
> [8] The Debate on RLVR Reasoning Capability Boundary: Shrinkage, Expansion, or Both? A Two-Stage Dynamic View. Arxiv 2025.
>
> [9] Deepseek-r1: Incentivizing reasoning capability in llms via reinforcement learning. Arxiv 2025.
>
> [10] DeepSeek-Prover-V2: Advancing Formal Mathematical Reasoning via Reinforcement Learning for Subgoal Decomposition. Arxiv 2025.
>
> [11] Let's verify step by step. ICLR 2024.

---

### Official Review · Reviewer_kTDb · 2025-10-25

**Soundness:** 2
**Presentation:** 3
**Contribution:** 3
**Rating:** 6
**Confidence:** 4

**Summary:**

This paper proposes TOKENSEEK, a plug-in for memory-efficient fine-tuning that is instance-aware at the token level. The core idea is to (1) seek salient tokens per input by combining context signals (attention-derived scores) and optimization signals (gradients of the loss w.r.t. penultimate-layer activations), then (2) ditch the rest by disabling gradient computation for unselected tokens during backprop (retaining forward activations but avoiding their gradient/optimizer memory).

**Strengths:**

Combines attention-derived context and gradient saliency to rank tokens per example; empirically more effective and more stable than random selection. Reports 2.8 GiB peak in one setting and ~15% of full-token QLoRA peak on Llama-3.2-1B, while maintaining accuracy; cumulative with PEFT (LoHa/QLoRA). isualizations show complementary early-token bias from attention and late-token focus from gradients; helps explain the chosen subset.

**Weaknesses:**

Need a controlled knob table for each baseline (checkpointing, offloading, micro-batching, seq length, optimizer sharding) and the resulting peak+average memory to ensure apples-to-apples comparisons. Gradient-based scoring requires a partial backward pass; quantify this overhead per step and analyze action oscillations/instability of the selected set across training. Provide seed variance tables.

**Questions:**

For LoRA/QLoRA/LoHa/IA3 and full FT, which memory knobs (checkpointing schedule, ZeRO, offload, seq length, grad accumulation, activation precision) were enabled? Please add a per-run configuration table with measured peak+average memory. How often does the selected token subset change across epochs/steps? Any evidence of training instabilities when the subset shifts? Could you learn α/β or the token fraction from validation loss via a small controller?

---

> ### Author Response · Authors · 2025-11-19
> **Author Response to Reviewer kTDb (Part 1/2)**
>
> Thank you for acknowledging the effectiveness, interpretability, generalizability, and empirical rigor of our method. We sincerely appreciate your time and effort in reviewing our paper and providing valuable comments. We provide explanations to your questions point-by-point in the following.
>
> ---
> ### **${\color{#f26921}\text{R1: Regarding the Knob Table for Each Baseline.}}$**
>
> > ${\color{#f26921}\text{Weakness-1:}}$ "Need a controlled knob table for each baseline (checkpointing, offloading, micro-batching, seq length, optimizer sharding) and the resulting peak+average memory to ensure apples-to-apples comparisons."
>
> ${\color{#f26921}\text{Response W1:}}$
>
> Thank you for your excellent suggestion regarding the training details. We fully agree about the importance of providing clearer training configurations for fair comparison and reproducibility.
>
> As revealed in Appendix S1, We train for one epoch using a learning rate of 4e−4 for all fine-tuning. A batch size of 1 is used with 32 gradient accumulation steps. Adapters are inserted into the feed-forward layers of each Transformer block following \[ref1\] and TokenTune (i.e., "gate\_proj", "down\_proj" and "up\_proj"). All sequence lengths are set to 2048\. We do not use any checkpointing, ZeRO or offloading techniques. Below are other training settings details.
>
> > bf16: true
> > fp16: false
> > lr\_scheduler\_type:"cosine"
> > warmup\_steps:100
> > weight\_decay:0.01
> > optim="adamw\_torch"
>
> In addition to the information already included in Appendix S1, we now provide the expanded configuration card in the format below, which has been added to Appendix S1.3 in the revised version.
>
> For QLoRA and LoHa, we use the below same configuration in all experiments.
>
> > alpha:16
> > dropout:0.05
> > r:8
>
> For other baselines such as IA3 and BOFT, we use the default PEFT configuration across all runs.
>
> Regarding the resulting peak / average memory, we’ve already reported the relative comparison in Table 1, and we would be happy to provide the original numbers for full parameter/token tuning in each setting for more comprehensive understanding. Other baseline memory usage can be calculated using the original numbers below and their relative memory percentages reported in Table 1\.
>
> |                | Average Memory (MB) | Maximum Memory (MB) |
> | -------------- | ------------------- | ------------------- |
> | Qwen2.5 (0.5B) | 12,070              | 27,300              |
> | Llama3.2 (1B)  | 19,462              | 39,688              |
> | Llama3.2 (3B)  | 40,100              | 83,927              |
>
> ---
> ### **${\color{#f26921}\text{R2: Regarding the Gradient-based Scoring.}}$**
>
> > ${\color{#f26921}\text{Weakness-2:}}$ "Gradient-based scoring requires a partial backward pass; quantify this overhead per step and analyze action oscillations/instability of the selected set across training. Provide seed variance tables."
>
> ${\color{#f26921}\text{Response W2:}}$
>
> Thank you for the excellent suggestion. Following your comments, we have added further quantification of our gradient scoring as summarized below.
>
> |                | Average Memory | Time (s) |
> | -------------- | ------------------- | -------- |
> | Qwen2.5 (0.5B) |         13.2%            |    291      |
> | Llama3.2 (1B)  |        11.5%             |     377     |
> | Llama3.2 (3B)  |         10.9%            |     566     |
>
>
> Storing 50% of token positions for Open-Platypus requires about 6.8 MB.
>
> Regarding the instability of the selected set across training, we have anlysed the stability of TokenSeek in Section 4.3, which show TokenSeek’s capability to maintain higher accuracy  with narrower fluctuations, emphasizing its robustness under varying memory constraints. Following your suggestion, we have added the variance tables below to improve the quality of our work.
>
> | 10%     | 20%     | 30%     | 40%     | 50%     |
> | ------- | ------- | ------- | ------- | ------- |
> | 0.05242 | 0.16229 | 0.01396 | 0.00669 | 0.01620 |
>
> ---
>
> **References:**
>
> \[1\] Towards a unified view of parameter-efficient transfer learning. arxiv 2021\.

---

> > ### Comment · Reviewer_kTDb · 2025-11-19
> >
> > Thank you so much for the prompt and detailed response
> > And the tables you added cleared my concerns regarding the config and overhaul.
> >
> > I just have a quick follow-up about the reproducibility and code release mentioned in the paper.
> > 1. Will the final repo include the full TokenSeek implementation, like token scoring/regrouping logic and the PEFT integration, or will it be more like a high-level execution script?
> > 2. It would be helpful to see a more specific pytorch style snippet or pseudocode for the token ditching
> > . Just to be clear, I'm not enforcing the early release of the code. I just want to gauge how easy it will be for the community to reimplementation TokenSeek based on the final camera-ready version.

---

> > > ### Author Response · Authors · 2025-11-19
> > >
> > > We truly appreciate your constructive engagement with our rebuttal. We’re glad our responses have addressed your concerns.
> > >
> > > 1. Yes, the final repository will include the **complete implementation** of TokenSeek, covering the scoring and regrouping, the full training pipeline, visualization utilities, and all PEFT integration components. Since our method is built on top of Huggingface PEFT, it can be directly combined with other PEFT techniques (even including those not covered in the paper) without additional engineering effort. We will also provide comprehensive training and evaluation scripts to support fully end-to-end reproduction. **We commit to incorporating all of these components into the final open-source release.**
> > >
> > >
> > > 2. We would also be pleased to share the PyTorch code used in our Llama experiments. Below is the forward implementation snippet of **LlamaDecoderLayer** that illustrates the regrouping and ditching.
> > >
> > >         residual = hidden_states
> > >         residual_prefix = hidden_states_prefix
> > >
> > >         with torch.no_grad():
> > >             hidden_states = self.input_layernorm(hidden_states)
> > >         hidden_states_prefix = self.input_layernorm(hidden_states_prefix)
> > >
> > >         # Self Attention
> > >         (
> > >             hidden_states,
> > >             hidden_states_prefix,
> > >             self_attn_weights,
> > >             present_key_value,
> > >         ) = self.self_attn(
> > >             hidden_states=hidden_states,
> > >             attention_mask=attention_mask,
> > >             position_ids=position_ids,
> > >             hidden_states_prefix=hidden_states_prefix,
> > >             attention_mask_prefix=attention_mask_prefix,
> > >             position_ids_prefix=position_ids_prefix,
> > >             past_key_value=past_key_value,
> > >             output_attentions=output_attentions,
> > >             use_cache=use_cache,
> > >         )
> > >         hidden_states = residual + hidden_states
> > >         hidden_states_prefix = residual_prefix + hidden_states_prefix
> > >
> > >         # Fully Connected
> > >         residual = hidden_states
> > >         residual_prefix = hidden_states_prefix
> > >
> > >         with torch.no_grad():
> > >             hidden_states = self.post_attention_layernorm(hidden_states)
> > >             hidden_states = self.mlp(hidden_states)
> > >             hidden_states = residual + hidden_states
> > >
> > >         hidden_states_prefix = self.post_attention_layernorm(hidden_states_prefix)
> > >         hidden_states_prefix = self.mlp(hidden_states_prefix)
> > >         hidden_states_prefix = residual_prefix + hidden_states_prefix
> > >
> > >         outputs = (
> > >             hidden_states,
> > >             hidden_states_prefix,
> > >         )
> > >
> > >         if output_attentions:
> > >             outputs += (self_attn_weights,)
> > >
> > >         if use_cache:
> > >             outputs += (present_key_value,)
> > >
> > >         return outputs
> > > We are always ready to respond to any further questions you may have, and we deeply thank you for your thoughtful support in improving our work.

---

> > > ### Author Response · Authors · 2025-11-27
> > > **Looking forward to the discussion**
> > >
> > > Dear Reviewer kTDb,
> > >
> > > We deeply appreciate the time and effort you’ve taken to review our work, especially given your busy schedule. As the authors-reviewer discussion phase draws to a close, we would be grateful to know whether we have adequately addressed your **follow-up concerns** and welcome any additional questions or points of discussion you'd like to raise.
> > >
> > > Thank you for your thoughtful consideration.
> > >
> > > Best regards,
> > >
> > > The Authors of Submission 11626

---

> ### Author Response · Authors · 2025-11-19
> **Author Response to Reviewer kTDb (Part 2/2)**
>
> ### **${\color{#f26921}\text{R3: Regarding the Memory Knobs for the Baseline.}}$**
>
> > ${\color{#f26921}\text{Question-1:}}$ "For LoRA/QLoRA/LoHa/IA3 and full FT, which memory knobs (checkpointing schedule, ZeRO, offload, seq length, grad accumulation, activation precision) were enabled? Please add a per-run configuration table with measured peak+average memory."
>
> ${\color{#f26921}\text{Response Q1:}}$ Please refer to our Response W1.
>
> ---
> ### **${\color{#f26921}\text{R4: Regarding the Selected Token Subset.}}$**
>
>
> > ${\color{#f26921}\text{Question-2:}}$ "How often does the selected token subset change across epochs/steps? Any evidence of training instabilities when the subset shifts?"
>
> ${\color{#f26921}\text{Response Q2:}}$
>
> Thank you for your insightful question. We would be happy to provide further clarifications here.
>
> TokenSeek operates in two stages: instance-aware token seeking and efficient token ditching. The former is performed **offline before training**, while the latter is applied **online during training**. Therefore, the selected token subset is **fixed** once the tunable percentage is determined, meaning it does not change across epochs or training steps.
>
> Regarding the instability of subset shifts, as discussed in Section 4.3, our scoring remains substantially more stable than random selection, which exhibits large variances. This difference underscores that token selection strategy matters: instance-aware TokenSeek yields far more consistent stability patterns than TokenTune’s random selection.
>
> ---
> ### **${\color{#f26921}\text{R5: Regarding the Learning of Hyperparameters.}}$**
>
> > ${\color{#f26921}\text{Question-3:}}$ "Could you learn α/β or the token fraction from validation loss via a small controller?"
>
> ${\color{#f26921}\text{Response Q3:}}$
>
> Thank you for raising this meaningful extension.
>
> Regarding the potential of learning α and β automatically, grid search over {\[1,0\], \[5,5\], \[7,3\], \[3,7\]} on a validation set is generally practical and sufficient. In our case, Open-Platypus lacks a validation split, and our ablation in Table 2 shows that performance remains stable across these settings, indicating low sensitivity within this range and limited marginal benefit from learning them externally.
>
> Regarding the potential of learning the token fraction **r** automatically, we clarify that **r is a resource controller**, which determines the number of tokens retained per sequence under a given memory budget. Our ablation in Figure 6 shows that memory decreases sharply as r moves from 50 percent down to 10 percent, making it more appropriate to *c*hoose r **based on the memory budget** rather than learn it via a single objective such as validation loss. A budget-driven choice of r better reflects the tradeoff between accuracy and memory savings.
>
> ---
>
> All modifications have been marked in ${\\color{\#f26921} orange}$ in our revised submission.
>
> **We appreciate your thoughtful comments. We hope our response addresses your concerns. Please let us know if there are any additional questions, and we will be happy to discuss further.**

---

### Official Review · Reviewer_CWhD · 2025-10-31

**Soundness:** 3
**Presentation:** 4
**Contribution:** 3
**Rating:** 8
**Confidence:** 4

**Summary:**

TOKENSEEK introduces a universally applicable plugin for memory-efficient fine-tuning of large language models without altering their architecture. It adaptively identifies and retains only the most salient tokens per instance—based on contextual and gradient importance—while discarding less useful ones to reduce activation memory. This instance-aware token ditching achieves significant memory savings with stable or improved performance, offering a generalizable, architecture-agnostic approach to efficient fine-tuning.

**Strengths:**

Solid problem formulation  as fine-tuning large LLMs is highly memory-intensive, with activations contributing a major share of the cost.
And Innovative approach to integrates gradient information with context scores to capture a more holistic measure of token importance, addressing the limitation that context-based evaluation alone reflects only intra-sequence relevance, not fine-tuning contribution.


Some of the specific strength:
 - Architecture-agnostic design: The proposed plugin can be applied to any pretrained LLM without modifying its core architecture that is very practical.
 - Strong empirical results: Demonstrates substantial memory savings while maintaining or improving model performance across benchmarks. Very solid results in table 1, specifically with QLora.
- Interpretability and analysis: Offers clear insights into token-level importance and provides comprehensive analysis on how token ditching affects fine-tuning. Figure 4 and other analysis plots are insightful.
- Clarity and presentation: This paper is very well-written, logically structured, and easy to follow, with clear motivation and experimental validation.

**Weaknesses:**

- Scalability concerns: The token regrouping step—where tokens are sorted by importance and selectively included for backpropagation—may pose significant implementation and communication challenges in large-scale distributed fine-tuning setups. Synchronizing token importance scores and managing uneven token partitions across devices could offset some of the claimed memory savings.

- Complexity of integration: Although conceptually modular, integrating the method into existing large-scale MEFT pipelines may require non-trivial modifications to data loading and parallelism strategies.

 - Limited large-scale validation: Experiments are mostly conducted on moderate-sized models (<=3B) and datasets; the method’s stability and efficiency under massive multi-node training scenarios remain unverified.

- Selective update imbalance: Dropping gradients for less salient tokens could bias training if token importance is misestimated or unstable across iterations.

**Questions:**

- In Eq 3, the notation of i and j is a bit confusing. It might be more clear to express context score for ${t_j}$ token as $\Sigma_{i=1}^{n} A_{ij}$. To clarify that you are summing the attention scores across all rows for the given column j.
- Similarly the notation in eq 4 should indicate that the the gradient score is for  token $t_j$.

- More clarification on the token regrouping process and its practicality in distributed fine-tuning settings would strengthen the paper.

---

> ### Author Response · Authors · 2025-11-19
> **Author Response to Reviewer CWhD (Part 1/3)**
>
> Thank you for acknowledging the novelty, methodological clarity, practicality, and empirical rigor of our method. We sincerely appreciate your time and effort in reviewing our paper and providing valuable comments. We provide explanations to your questions point-by-point in the following.
>
> ---
> ### **${\color{#f26921}\text{R1: Regarding the Scalability Concerns.}}$**
>
> > ${\color{#f26921}\text{Weakness-1:}}$ "The token regrouping step—where tokens are sorted by importance and selectively included for backpropagation—may pose significant implementation and communication challenges in large-scale distributed fine-tuning setups. Synchronizing token importance scores and managing uneven token partitions across devices could offset some of the claimed memory savings."
>
> ${\color{#f26921}\text{Response W1:}}$
>
> Thank you for this insightful question, which significantly deepens our work. We would be happy to further discuss its extension to larger and more complex distributed environments.
>
> We consider two major distributed fine-tuning settings: (1) **DP:** data-parallel training (including ZeRO/FSDP variants), and (2) **TP/SP:** tensor/sequence parallelism (model parallel training).
>
> - For DP, each GPU holds a full copy of the model and only processes a different batch. Gradients are all-reduced at the end. DP has minimal impact on TokenSeek because every GPU handles its own local samples independently and only needs to synchronize parameters, not token-level information.
>
> - For TP/SP, we would like to further analyze it separately since TokenSeek operates in two stages: instance-aware token seeking and efficient token ditching. The former is performed **offline** before training, while the latter is applied **online** during training.
>
> 	- For scoring, we involve the calculation of gradient score, which is computed only at the penultimate layer (all earlier layers are frozen), requiring substantially less memory than full fine-tuning (i.e., requires only **13.2% memory** of full fine tuning under qwen settings). It is inevitable to communicate gradients across GPUs to assemble the final gradient. However, because we compute gradients only in the penultimate layer, the additional memory cost remains manageable compared with full fine-tuning.
> 	- For training, we can distribute the token-score dictionary for each instance across GPUs before training, which consumes only minimal memory to store the mapping (e.g., storing 50% of token positions for Open-Platypus requires about **6.8 MB**). During training, this regrouping and reorganizing introduces an extra communication step for handling irregular gradient computation. However, thanks to efficient token ditching, the amount of gradient that needs to be synchronized is greatly reduced.
>
> In conclusion, we provide a preliminary analysis of our **unoptimized plain implementation** and the communication challenges of applying TokenSeek in distributed environments. While these factors may reduce some of the memory savings observed in single-node training, the two-stage design combining partial-gradient scoring and partial-gradient updating keeps the overhead **controllable** compared with full fine-tuning.
>
> We are also looking forward to **collaborating with extraordinary engineering teams to further optimize** TokenSeek for more complex large-scale training scenarios.
>
> ---
> ### **${\color{#f26921}\text{R2: Regarding the Complexity of Integration.}}$**
>
> > ${\color{#f26921}\text{Weakness-2:}}$ "Although conceptually modular, integrating the method into existing large-scale MEFT pipelines may require non-trivial modifications to data loading and parallelism strategies."
>
> ${\color{#f26921}\text{Response W2:}}$
>
> Thank you for this practical concern. Our implementation is based on huggingface’s transformers and PEFT, which allows a single integration on one model to be directly **reused and combined** with other PEFT methods. Our code will also be released upon acceptance.
>
> Specifically, we provide a detailed explanation of the modifications we make to each model below. As described in Appendix S1, we regroup the input I into \[I\_selected, I\_unselected\], apply “torch.no\_grad()” to all I\_unselected, and finally reorganize \[O\_selected, O\_unselected\] into the output O. This procedure is **model-agnostic**, follows a **common pattern**, and does not require manual adaptation to different model architectures, which can be handled by code agents that are highly capable of capturing these patterns, making the extension to other models straightforward.

---

> ### Author Response · Authors · 2025-11-19
> **Author Response to Reviewer CWhD (Part 2/3)**
>
> ### **${\color{#f26921}\text{R3: Regarding the Large-scale Validation.}}$**
>
>
> > ${\color{#f26921}\text{Weakness-3:}}$  "Experiments are mostly conducted on moderate-sized models (\<=3B) and datasets; the method’s stability and efficiency under massive multi-node training scenarios remain unverified."
>
> ${\color{#f26921}\text{Response W3:}}$
>
> This is an excellent question, and we fully agree with the reviewer that validating on larger-scale models is necessary.
>
> As shown in Appendix S3, we conducted a preliminary experiment on the Llama2 7B model Specifically, under full-token tuning shown in Table S3, TokenSeek achieves an average score of 60.97, slightly improving over baseline (60.73), while reducing both average and peak memory by 22.1% and 54.4%, respectively. More notably, under the QLoRA setting, TokenSeek attains the comparable performance (62.14) while requiring only 18.2% average memory and 12.4% peak memory compared to full tuning, demonstrating a \>80% memory saving. These results validate the scalability and robustness of our method even in large-model scenarios, reinforcing its value in memory-constrained training environments.
>
> ---
> ### **${\color{#f26921}\text{R4: Regarding the Selective Update Imbalance.}}$**
>
> > ${\color{#f26921}\text{Weakness-4:}}$  "Dropping gradients for less salient tokens could bias training if token importance is misestimated or unstable across iterations."
>
> ${\color{#f26921}\text{Response W4:}}$
>
> Thank you for raising this concern, and we would like to provide further clarification here.
>
> 1. We retain full-sequence attention and loss computation in the forward pass, and only zero out gradients for unselected tokens during backpropagation. This keeps the training objective and **context intact** while updating only the gradients deemed most important. This approach constitutes structured gradient sparsification rather than sample dropping or parameter pruning, and **all parameters are still updated at every step**, reducing the risk of systematic bias.
> 2. Beyond combining contextual and gradient-based signals to reduce potential misestimation from any single indicator, the scoring is derived from the current sample. It is therefore an **instance-aware, dynamically updated selection** mechanism rather than a fixed rule. Even if some iterations introduce noise, subsequent training iterations across many examples will adaptively mitigate it.
>
> This two scoring and training designs enable TokenSeek to achieve stable evaluation (see mode details in the section 4.3), align with the empirical results from Figure 3\.
>
> Furthermore, we also investigate the influences of misestimating token importance. Under the Llama3.2 (1B) QLoRA setup, we introduce an additional setting that selects Top 10% plus Top 40–50% tokens, instead of the standard Top 20%, to simulate misestimation.
>
> | Settings             | Overall  Tunable Token | MMLU  | ARC   | Hella Swag | Truthful QA | Wino Grande | Average Score |
> | -------------------- | ---------------------- | ----- | ----- | ---------- | ----------- | ----------- | ------------- |
> | Random 20%           | 20%                    | 40.10 | 63.93 | 42.96      | 43.23       | 61.01       | 50.25         |
> | Top 20%              | 20%                    | 39.42 | 65.73 | 53.24      | 39.86       | 60.77       | 51.80         |
> | Top 20% + Top 40-50% | 20%                    | 39.16 | 65.91 | 51.20      | 39.28       | 61.01       | 51.31         |
>
> Although “Top 10% \+ Top 40-50%” underperforms “Top 20%,” it still outperforms “Random 20%,” demonstrating the robustness of TokenSeek.
>
> In the future, we plan to explore whether smoothing the scoring function or injecting a small portion of randomly selected tokens as exploration can further improve TokenSeek.
>
> ---
> ### **${\color{#f26921}\text{R5: Regarding the Notations in Equation 3 and 4.}}$**
>
> > ${\color{#f26921}\text{Question-1:}}$ "In Eq 3, the notation …, Similarly the notation in eq 4 should indicate that the the gradient score is for token .."
>
> ${\color{#f26921}\text{Response Q1:}}$
>
> Sorry for the confusion. We sincerely appreciate your careful reading and for pointing this out, we have revised accordingly to ensure clarity and precision.

---

> ### Author Response · Authors · 2025-11-19
> **Author Response to Reviewer CWhD (Part 3/3)**
>
> ### **${\color{#f26921}\text{R6: Regarding the Clarification on the Token Regrouping Process.}}$**
>
> > ${\color{#f26921}\text{Question-2:}}$ "More clarification on the token regrouping process and its practicality in distributed fine-tuning settings would strengthen the paper."
>
> ${\color{#f26921}\text{Response Q2:}}$
>
> Thank you again for raising this meaningful question. Beyond the discussion provided in Appendix S1 and Response W1, we further elaborate on the regrouping process in distributed environments.
>
> In tensor parallelism, hidden dimensions are split across GPUs, so token regrouping is purely a local row reindexing operation. TP’s usual all-reduce pattern stays unchanged. In sequence parallelism, however, the sequence dimension is sharded, so splitting tokens into selected and unselected sets breaks the local-contiguous token assumption. Each layer therefore requires an all-to-all shuffle to regroup tokens back to their original global order before proceeding. As a result, SP introduces small but necessary per-layer communication for token restoration.
>
> ---
>
> All modifications have been marked in ${\\color{\#f26921} orange}$ in our revised submission.
>
> **We appreciate your thoughtful comments. We hope our response addresses your concerns. Please let us know if there are any additional questions, and we will be happy to discuss further.**

---

### Official Review · Reviewer_ZpqT · 2025-11-05

**Soundness:** 3
**Presentation:** 3
**Contribution:** 2
**Rating:** 4
**Confidence:** 3

**Summary:**

This paper addresses the limitation of current fine-tuning methods that largely overlook the information contained in individual training instances. The authors propose TokenSeek, a universal plug-in framework for Transformer-based models that performs instance-aware token selection and ditching. By selectively fine-tuning on important tokens identified via attention and gradient signals, TokenSeek reduces activation memory consumption while maintaining comparable performance.

**Strengths:**

- The paper is clearly written and generally easy to follow, with only minor typos.
- The proposed TokenSeek method is conceptually simple and practically implementable.
- Experimental results demonstrate competitive performance on multiple downstream tasks (e.g., QA, reasoning) across different LLM architectures such as LLaMA and Qwen.
- The experiments are comprehensive and include ablations and cross-task evaluations, which strengthen empirical credibility.

**Weaknesses:**

- Limited Novelty Clarification:
The main contribution—token importance–based selection—extends the previous TokenTune framework rather than introducing a fully new paradigm. While TokenSeek improves token importance estimation compared to random selection, the core ideas of memory reduction and generalizability are largely inherited from TokenTune. The paper should better articulate what is fundamentally novel about TokenSeek beyond methodological refinements.

- Comparison to Low-Rank and Partial-Tuning Methods:
On Line 835, the paper notes that TokenSeek and TokenTune incur ~11–15% more GPU hours than full-token tuning. In contrast, low-rank PEFT methods (e.g., LoRA) typically achieve both lower memory and faster training. Since TokenSeek inherits TokenTune’s extra computational overhead, the paper should clarify whether TokenSeek offers any non-trivial advantages over low-rank or partial-tuning approaches.

- Missing Baselines and Related Work:
Several recent sparsity-based PEFT methods are missing from the related work and experimental comparisons. As TokenSeek belongs to the PEFT family, adding one representative partial-tuning or sparse fine-tuning baseline (e.g., [1–3]) would provide a more complete evaluation and contextualization of the method’s contribution.


- Some minor typos
“as showin in Fig. 1 (a)” → shown;
“Benifit from” → Benefit from;
L313 “Unde” → Under;
“achiving” → achieving;
“TokenSeek achieve” → achieves;
“LoRA/QLoRA achieves” → plural → achieve




[1] Scaling Sparse Fine-Tuning to Large Language Models

[2] Sparse Matrix in Large Language Model Fine-tuning

[3] The Lottery Ticket Hypothesis: Finding Sparse, Trainable Neural Networks

**Questions:**

- Code-Domain Generalization:
The experiments cover QA and Open-Platypus datasets. How would TokenSeek perform on code-related datasets (e.g., CodeAlpaca, HumanEval)? Code data tends to have dense information where most tokens are important, which could reduce token sparsity efficiency. Including one experiment on a code dataset would better demonstrate TokenSeek’s generalizability.

- Source of Memory Savings:
Please clarify precisely where the memory savings originate. Since token selection is gradient-based, forward propagation must still process all tokens, and optimizer memory (e.g., Adam states) typically depends on trainable parameters rather than token count. How does TokenSeek achieve the reported activation memory reduction—through selective gradient storage, reduced backward pass, or another mechanism? A detailed explanation would strengthen the technical soundness.



At present, this appears to be a borderline paper (score ~5). The core idea is promising, but its novelty and advantage over prior TokenTune and LoRA-based methods need clearer articulation. If the authors can convincingly address the questions above—particularly regarding memory savings and broader generalization—I would be open to raising my score.

---

> ### Author Response · Authors · 2025-11-19
> **Author Response to Reviewer ZpqT (Part 1/3)**
>
> Thank you for acknowledging the methodological clarity, practicality, and empirical rigor of our method. We sincerely appreciate your time and effort in reviewing our paper and providing valuable comments. We provide explanations to your questions point-by-point in the following.
>
> ---
> ### **${\color{#f26921}\text{R1: Regarding the Novelty Clarification.}}$**
>
> Thank you for highlighting this important point about clarifying our novelty. We appreciate the opportunity to further articulate the novelty of TokenSeek and its distinctions from TokenTune.
>
> > ${\color{#f26921}\text{Weakness-1:}}$ "The main contribution—token importance–based selection—extends the previous TokenTune framework rather than introducing a fully new paradigm. ……. The paper should better articulate what is fundamentally novel about TokenSeek beyond methodological refinements."
>
> ${\color{#f26921}\text{Response W1:}}$
>
> Although both methods aim to improve memory efficiency in fine-tuning, their underlying **motivations**, **scoring mechanisms**, and **empirical behaviors** differ fundamentally. TokenTune relies on data-agnostic partial-gradient selection or random token dropping inspired by an engineering perspective, which we found to be **ineffective and unstable** across tasks. In contrast, TokenSeek is motivated by the observation that not all tokens contribute equally to model updates, and therefore adopts a data-driven, instance-aware criterion.
>
> Specifically, TokenSeek introduces a hybrid scoring mechanism that integrates both attention-based contextual relevance and gradient-based optimization signals. This leads to substantially improved stability, interpretability, and effectiveness compared to random or data-agnostic selection. Beyond memory and performance metrics, TokenSeek also incorporates **comprehensive analyses on stability, interpretability, and optimization behavior**, offering insights into token-level contribution during efficient fine-tuning.
>
> Finally, to the best of our knowledge, no prior MEFT method performs instance-aware token selection for activation-memory–efficient training. This instance-aware perspective represents the core novelty of TokenSeek and distinguishes it from TokenTune’s engineering-oriented design.
>
> > ${\color{#f26921}\text{Weakness-2:}}$  "While TokenSeek improves token importance estimation compared to random selection, the core ideas of memory reduction and generalizability are largely inherited from TokenTune."
>
> ${\color{#f26921}\text{Response W2:}}$
>
> While TokenSeek adopts the same high-level token-ditching paradigm as TokenTune, its advantages are **substantially amplified** due to our data-driven scoring design.
>
> Memory reduction.
> TokenTune’s random dropping causes unstable and degraded performance, forcing it to keep more tokens to stay competitive. In contrast, TokenSeek identifies truly salient tokens, enabling us to discard far more activations without hurting accuracy. As a result, under equal performance, TokenSeek consistently **achieves significantly lower memory** (i.e., with only 10 percent tunable tokens, we achieve even higher performance than TokenTune’s 50 percent setting under the same model configuration, as shown in Figure 3).
>
> Generalizability.
> TokenSeek relies solely on inherent signals from the pretrained model (attention and gradients). It requires **no auxiliary model, no task-specific knowledge, and no architectural modification**. This makes it compatible with any Transformer-based model and any PEFT method. In contrast, several recent MEFT paradigms depend on customized modules (e.g., reversible networks), which limits their applicability. TokenSeek remains universally plug-and-play across architectures and domains.
>
> ---
> ### **${\color{#f26921}\text{R2: Regarding the Comparison to Low-Rank and Partial-Tuning Methods.}}$**
>
> > ${\color{#f26921}\text{Weakness-3:}}$  "On Line 835, the paper notes that TokenSeek and TokenTune incur \~11–15% more GPU hours than full-token tuning. In contrast, low-rank PEFT methods (e.g., LoRA) typically achieve both lower memory and faster training. Since TokenSeek inherits TokenTune’s extra computational overhead, the paper should clarify whether TokenSeek offers any non-trivial advantages over low-rank or partial-tuning approaches."
>
> ${\color{#f26921}\text{Response W3:}}$
>
> Thank you for the insightful observation regarding the baseline comparison. We would like to provide a clearer explanation here.
>
> While LoRA may train faster, TokenSeek provides substantially **better memory efficiency** and **more stable performance**. Under the Qwen setting in Table 1, TokenSeek achieves both lower memory usage (19.2% vs. 81.2%) and higher accuracy (48.45 vs. 48.06) compared with LoRA. Notably, TokenTune exhibits clear performance degradation (46.34 vs. 48.06) in this comparison.
>
> We have revised our writing and highlighted these advantages in our revision.

---

> ### Author Response · Authors · 2025-11-19
> **Author Response to Reviewer ZpqT (Part 2/3)**
>
> ### **${\color{#f26921}\text{R3: Regarding the Baselines and Related Work.}}$**
>
> > ${\color{#f26921}\text{Weakness-4:}}$  "Several recent sparsity-based PEFT methods are missing from the related work and experimental comparisons. As TokenSeek belongs to the PEFT family, adding one representative partial-tuning or sparse fine-tuning baseline (e.g., \[1–3\]) would provide a more complete evaluation and contextualization of the method’s contribution."
>
> ${\color{#f26921}\text{Response W4:}}$
>
> Thank you for the excellent suggestion. We completely agree with you about the diversity of baselines we should incorporate for comprehensive comparison.
>
> We have included \[ref1-3\] in our related work discussion. In our previous submission, we included **BOFT** (ICLR 2024, officially merged into Huggingface PEFT) as a **representative sparsity-based PEFT method in Table 1**. Following your comments, we conducted additional experiments on RanLoRA \[ref4\] (ICLR 2025, also officially merged into Huggingface PEFT) under the Qwen2.5 (0.5B) setting, as summarized below.
>
> | Method                | Ave. Mem. | Max. Mem. | ARC   | HellaSwag | MMLU  | TruthfulQA | WinoGrande | Average Score |
> | --------------------- | --------- | --------- | ----- | --------- | ----- | ---------- | ---------- | ------------- |
> | BOFT                  | 145.1%    | 100.6%    | 34.64 | 51.70     | 58.18 | 39.57      | 56.43      | 48.10         |
> | RanLoRA               | 95.4%     | 86.7%     | 29.18 | 50.10     | 58.33 | 45.21      | 57.22      | 48.01         |
> | QLoRA                 | 51.7%     | 45.6%     | 34.64 | 50.10     | 58.05 | 40.41      | 55.09      | 47.66         |
> | - w/ TokenSeek (Ours) | 19.2%     | 13.4%     | 34.56 | 50.09     | 57.52 | 41.51      | 58.56      | **48.45**     |
>
> These results further enhance the comprehensiveness of our comparison and greatly deepen our paper demonstrate the effectiveness of our approach under the requested setting. We have included the new experiments and corresponding discussion in the revised version.
>
> ---
> ### **${\color{#f26921}\text{R4: Regarding the Typos.}}$**
>
> > ${\color{#f26921}\text{Weakness-5:}}$ "Some minor typos “as showin in Fig. 1 (a)” → shown; “Benifit from” → Benefit from; L313 “Unde” → Under; “achiving” → achieving; “TokenSeek achieve” → achieves; “LoRA/QLoRA achieves” → plural → achieve"
>
> ${\color{#f26921}\text{Response W5:}}$
>
> We sincerely appreciate your careful reading and for pointing this out, we have revised accordingly to ensure clarity and precision.
>
> ---
>
> **References:**
>
> \[1\] Scaling Sparse Fine-Tuning to Large Language Models. arxiv 2024\.
>
> \[2\] Sparse Matrix in Large Language Model Fine-tuning. arxiv 2024\.
>
> \[3\] The Lottery Ticket Hypothesis: Finding Sparse, Trainable Neural Networks. ICLR 2019\.
>
> \[4\] RandLora: Full-rank parameter-efficient fine-tuning of large models. ICLR 2025\.

---

> ### Author Response · Authors · 2025-11-19
> **Author Response to Reviewer ZpqT (Part 3/3)**
>
> ---
> ### **${\color{#f26921}\text{R5: Regarding the Code-Domain Generalization.}}$**
>
> > ${\color{#f26921}\text{Question-1:}}$ "The experiments cover QA and Open-Platypus datasets. How would TokenSeek perform on code-related datasets (e.g., CodeAlpaca, HumanEval)? Code data tends to have dense information where most tokens are important, which could reduce token sparsity efficiency. Including one experiment on a code dataset would better demonstrate TokenSeek’s generalizability."
>
> ${\color{#f26921}\text{Response Q1:}}$
>
> We especially thank you for your constructive suggestions, which significantly improve the quality of our paper. We truly agree that evaluating on coding tasks is important, and we have conducted preliminary experiments under the Llama3.2 (1B) setting as follows.
>
> | Method                  | Ave. Mem. | Max. Mem. | ARC   | HellaSwag | MMLU  | TruthfulQA | WinoGrande | Humaneval | Average Score |
> | ----------------------- | --------- | --------- | ----- | --------- | ----- | ---------- | ---------- | --------- | ------------- |
> | **LoHa**                | 92.3%     | 99.4%     | 39.25 | 65.93     | 57.60 | 37.87      | 60.77      | 13.41     | 45.81         |
> | – w/ TokenTune (Random) | 45.9%     | 28.4%     | 38.48 | 64.21     | 50.34 | 43.89      | 59.91      | 10.97     | 44.63         |
> | – w/ TokenSeek (Ours)   | 45.9%     | 28.4%     | 38.57 | 65.89     | 58.18 | 39.34      | 60.93      | 14.02     | 46.16         |
> | **QLoRA**               | 45.6%     | 34.8%     | 38.82 | 65.26     | 56.39 | 38.85      | 61.33      | 14.02     | 45.78         |
> | – w/ TokenTune (Random) | 14.8%     | 14.3%     | 39.33 | 62.97     | 41.76 | 41.36      | 60.69      | 12.80     | 43.15         |
> | – w/ TokenSeek (Ours)   | 14.8%     | 14.3%     | 39.08 | 65.98     | 58.03 | 38.65      | 61.33      | 14.63     | 46.28         |
>
> Although coding tasks may contain denser information than QA tasks, TokenSeek still performs effectively, which may be because of our instance-aware token ditching and the strategy that we preserve the full forward pass, allowing complete attention and contextual information to remain intact.
>
> We have included the new experiments and the corresponding discussion in the revised version.
>
> ---
> ### **${\color{#f26921}\text{R6: Regarding the Source of Memory Savings.}}$**
>
> > ${\color{#f26921}\text{Question-2:}}$  "Please clarify precisely where the memory savings originate. Since token selection is gradient-based, forward propagation must still process all tokens, and optimizer memory (e.g., Adam states) typically depends on trainable parameters rather than token count. How does TokenSeek achieve the reported activation memory reduction—through selective gradient storage, reduced backward pass, or another mechanism? A detailed explanation would strengthen the technical soundness."
>
> ${\color{#f26921}\text{Response Q2:}}$
>
> We would like to further clarify the source of memory savings. TokenSeek operates in two stages: instance-aware token seeking and efficient token ditching. The former is performed **offline before training**, while the latter is applied **online during training**.
>
> Specifically, for token seeking, we combine context-based and gradient-based scores. The context score is obtained through a forward pass, and the gradient score is computed only at the **penultimate layer** (all earlier layers are frozen). Both scores require substantially less memory than full fine-tuning (i.e., requires only 13.2% memory of full fine tuning under qwen settings). After scoring, we sort tokens and store each instance’s mapping for subsequent training, where gradients for unselected tokens are ditched based on the stored mapping. As detailed in Sections 3.1 and 3.2.2, tuning only 10% of tokens theoretically needs about \~1% of the activation memory, and our empirical results on Llama-3.2-1B (2.8 GB, 14.8% of the original memory) align with this theoretical analysis.
>
> Overall, the memory savings arise from our **partial backward scoring mechanism** and the **instance-aware direct token ditching strategy**. We have refined the corresponding writing and added further discussion in the revision.
>
> ---
>
> All modifications have been marked in ${\\color{\#f26921} orange}$ in our revised submission.
>
> **We appreciate your thoughtful comments. We hope our response addresses your concerns. Please let us know if there are any additional questions, and we will be happy to discuss further.**

---

> ### Author Response · Authors · 2025-11-27
> **Looking forward to the discussion**
>
> Dear Reviewer ZpqT,
>
> We deeply appreciate the time and effort you’ve taken to review our work, especially given your busy schedule. As the authors-reviewer discussion phase draws to a close, we would be grateful for the opportunity to engage in dialogue with you. Our goal is to ensure we've adequately addressed your concerns and welcome any additional questions or points of discussion you'd like to raise.
>
>
> Thank you for your thoughtful consideration.
>
>
> Best regards,
>
> The Authors of Submission 11626

---

### Author Response · Authors · 2025-12-01
**Final Remarks by Authors of Submission 11626**

## I. Acknowledgments

We would like to express our sincere gratitude to all reviewers for their insightful comments and constructive suggestions. We especially appreciate Reviewer `kTDb` and `cH6y` for their thoughtful consideration and constructive engagement with our rebuttal - it has significantly strengthened our work.

**${\color{#f26921}\text{Before the discussion}}$**, we appreciate the recognition from Reviewer CWhD (Rating: 8) and kTDb (Rating: 6), as well as the positive-leaning approval from Reviewer ZpqT (Rating: 4) and cH6y (Rating: 4), both of whom explicitly expressed a willingness to **reconsider their rating** (e.g., `“I would be open to raising my score”` from Reviewer ZpqT and `“I would reconsider my score if these concerns are adequately addressed”` from Reviewer cH6y).

**${\color{#f26921}\text{During the discussion}}$**, we are pleased that the major concerns raised by Reviewer kTDb and cH6y were resolved through our additional clarifications and results (e.g., `“the tables you added cleared my concerns regarding the config and overhaul”` from Reviewer kTDb and `“Many of my earlier concerns have been addressed”` from Reviewer cH6y). We especially appreciate Reviewer cH6y’s **inclination to increase the score** from 4 to 6 on 26 Nov 2025, 03:25 ET.

---

## II. Key Strengths

Reviewers highlighted strengths across five dimensions:

- **Novelty and Innovation**
  - Instance-aware paradigm combining attention and gradient signals (CWhD, kTDb)
  - Insight that training tokens contribute unequally to fine-tuning (cH6y)
  - Simple yet effective improvement over random selection (ZpqT)
- **Effectiveness and Efficiency**
  - Large memory savings (e.g., 2.8 GiB peak on Llama-3.2-1B) with preserved accuracy (kTDb, CWhD)
  - Competitive performance across QA and reasoning tasks and architectures (ZpqT)
  - Outperforms QLoRA while using significantly less memory (CWhD)
- **Practicality**
  - Architecture-agnostic, plug-and-play design without model changes (CWhD)
  - Straightforward to implement (ZpqT)
- **Interpretability**
  - Transparent scoring mechanism revealing token importance (cH6y, CWhD)
  - Clear visualizations of attention and gradient bias (cH6y, kTDb)
- **Methodological Clarity and Empirical Rigor**
  - Clear writing, solid formulation, generally easy to follow (ZpqT, CWhD, cH6y)
  - Comprehensive experiments with ablations and cross-task evaluations; strong motivation and empirical validation, well-organized presentation (ZpqT, CWhD, kTDb)

------

## III. Key Concerns and Our Responses

| **Key Concerns**                                             | **Reviewers** | **Our Response**                                             |
| ------------------------------------------------------------ | ------------- | ------------------------------------------------------------ |
| *Novelty vs TokenTune.* Concern that TokenSeek extends TokenTune without a new paradigm. | ZpqT          | TokenSeek uses a data-driven, instance-aware mechanism, fundamentally different from TokenTune’s unstable random and data-agnostic approach. |
| *Baselines & Generalization.* Need more sparse PEFT baselines and code experiments. | ZpqT, kTDb    | Besides BOFT **in original paper**, added RanLoRA baselines, evaluated on HumanEval, and provided detailed training knob tables. |
| *Memory Savings Mechanism.* How gradient-based scoring reduces memory despite backward pass. | ZpqT, kTDb    | Clarified two-stage design: offline partial-backward scoring on the penultimate layer using ~11% memory, then online token ditching. |
| *Scalability & Distributed Training.* Challenges in regrouping and communication. | CWhD          | Analyzed DP and TP/SP settings, showing controllable overhead. Validated scalability on Llama2-7B **in original paper** with over 80 percent memory savings and comparable performance. |
| *Hyperparameters & Stability.* Need justification for α and β choices and subset-shift stability. | cH6y, kTDb    | Ablations **in original paper** show low sensitivity. Subsets are fixed after scoring, and variance tables confirm stability. |
| *“Global Anchor” Interpretation.* Why specific tokens are consistently preserved. | cH6y          | Explained that attention sinks naturally receive high combined scores, leading TokenSeek to preserve them without artificial bias. |

---

## IV. Commitment to Revision

We have already incorporated all discussion points and additional experiments into our revision. All modifications have been marked in ${\\color{\#f26921} orange}$ in our revised submission. This includes adding the new experimental results on RanLoRA and HumanEval, the detailed training configuration tables, and the deeper discussions on "global anchors" and reasoning domain connections. We are also committed to continually incorporating the feedback from the discussion into the revision to polish our work.

------

**We deeply appreciate the expertise and time of the AC and reviewers.**

---

### Meta-Review · Area_Chair_K4jF · 2025-12-17

**Summary:**

This paper prompted substantial discussion around the novelty relative to prior token-dropping and PEFT methods, the precise source of memory savings, and the practicality of deploying instance-aware token regrouping in large-scale or distributed training settings.

Reviewers generally agreed on the core idea: instance-aware token selection using combined attention and gradient signals. But they initially questioned whether it went beyond an incremental refinement of existing approaches. The rebuttal managed to clarify the mechanism, provided missing baselines and configuration details, and strengthened empirical support across additional tasks and settings.

Overall, while some positioning and scope questions remain, the discussion converged toward recognizing the method as a meaningful contribution to memory-efficient fine-tuning.

**Reviewer Concerns:**

I believe most major technical concerns were addressed during the rebuttal. For example, requests for clearer explanation of where memory savings arise, more complete baseline comparisons, controlled configuration tables, stability and variance analysis, and broader task coverage (including code domains) were handled.

Concerns about HP sensitivity, gradient-layer choice, and interpretability of preserved tokens were also largely resolved through ablations and analysis.

One remaining concern is how to more deeply situate the token-importance findings relative to parallel observations in reasoning-focused RL/SFT literature: this remains partially open, but is more about framing and depth of analysis than correctness or validity.

**Reviewer Scores:**

Based on the discussion trajectory, my educated guess is that reviewers who were initially borderline negative or uncertain would likely have raised their scores into the weak accept range after the rebuttal. The score distribution would plausibly shift upward from mixed borderline evaluations to a modest but consistent acceptance signal, although I don't think the enthusiasm will be huge.

---

### Decision · Program_Chairs · 2026-01-26

Accept (Poster)